# Parsel🐍: Algorithmic Reasoning with Language Models by Composing Decompositions

**Eric Zelikman, Qian Huang, Gabriel Poesia, Noah D. Goodman, Nick Haber**
Stanford University
{ezelikman, qhwang, poesia, ngoodman, nhaber}@stanford.edu

## Abstract

Despite recent success in large language model (LLM) reasoning, LLMs struggle with hierarchical multi-step reasoning tasks like generating complex programs. For these tasks, humans often start with a high-level algorithmic design and implement each part gradually. We introduce Parsel, a framework enabling automatic implementation and validation of complex algorithms with code LLMs. With Parsel, we automatically decompose algorithmic tasks into hierarchical natural language function descriptions and then search over combinations of possible function implementations using tests. We show that Parsel can be used across domains requiring hierarchical reasoning, including program synthesis and robotic planning. We find that, using Parsel, LLMs solve more competition-level problems in the APPS dataset, resulting in pass rates over 75% higher than prior results from directly sampling AlphaCode and Codex, while often using a smaller sample budget. Moreover, with automatically generated tests, we find that Parsel can improve the state-of-the-art pass@1 performance on HumanEval from 67% to 85%. We also find that LLM-generated robotic plans using Parsel are more than twice as likely to be considered accurate than directly generated plans. Lastly, we explore how Parsel addresses LLM limitations and discuss how Parsel may be useful for human programmers. We release our code at https://github.com/ezelikman/parsel.

## 1 Introduction

To a language model for code (as for a human), each new token is a new chance to break the program. Chen et al. [12] highlighted this issue in a simple synthetic experiment: when asked to generate a program with a series of simple string transformations, the performance of their code language model, Codex, drops dramatically with the number of steps. As they pointed out, a human who can implement a few building blocks should be able to compose these blocks with arbitrary length.

Unlike other token generators, human programmers have (mostly) learned to break down complex tasks into manageable parts that work alone (i.e., are modular) and work together (i.e., are compositional). And, when human-generated tokens cause functions to break, the functions can ideally be rewritten independently of the rest of the program. In contrast, naively, we expect code language models to generate token sequences that are correct in their entirety. Motivated by this, we study how to leverage language models to decompose problems and assemble their compositional solutions.

We introduce Parsel, a framework that enables the decomposition, implementation, then composition of complex programs from natural language. The process consists of three main steps: 1) A language model generates a Parsel program by decomposing a natural language task into natural language function descriptions. 2) For each function description, a language model generates modular implementations. 3) The Parsel synthesizer then builds up programs by testing minimal, combinatorial groups of implementations against sets of constraints (e.g. input-output examples).

37th Conference on Neural Information Processing Systems (NeurIPS 2023).

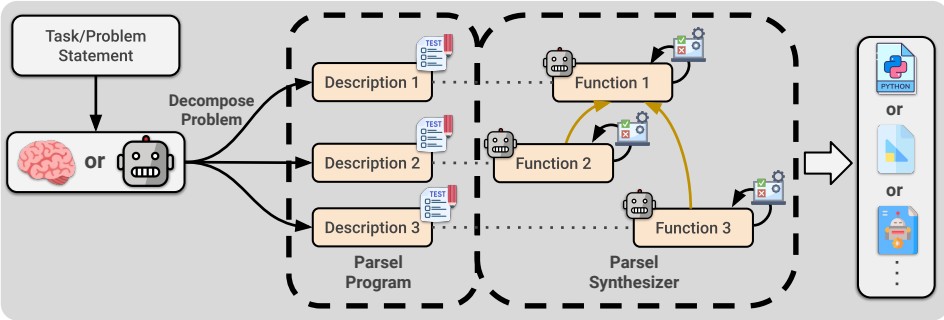

Figure 1: **Parsel overview**. A human or LLM writes a task decomposition (in Parsel), which is split into its strongly-connected components (SCC), and then the Parsel synthesizer uses a code LLM and a constraint solver to implement and compose each SCC. When all functions have constraints and there is no recursion, each function is its own SCC. We provide a more detailed figure in Appendix H.

Thus, generating and implementing Parsel mirrors a pattern in human reasoning – decomposing an abstract plan until it can be solved automatically [55] – and this compositional structure also benefits language models. We show that LLMs can generate Parsel with only few-shot prompting. Using their solutions as input to the Parsel synthesizer, we outperform prior work on competition-level problems in the APPS dataset [27], including AlphaCode [35] and both versions of Codex [12, 11]. We also show that with GPT-4 [44], not only can we generate Parsel *zero-shot* just by describing its format, but this allows it to substantially outperform GPT-4 alone on HumanEval [12], with and without generated tests. By automatically generating and implementing Parsel, LLMs also propose robotic plans that are more accurate than in prior work on language models as robotic planners [28]. Broadly, we formulate Parsel as a general-purpose algorithmic reasoning framework.

## 2 Methods

In this section, we provide an overview of the Parsel framework. First, we show how programs can be specified in Parsel. Afterward, we explain how language models can generate Parsel from task descriptions. Finally, we present the Parsel synthesizer that allows us to actually implement Parsel programs into runnable code. Specifically, we further discuss how we handle various challenges in solving the constraints, from recursion to test underspecification. There are several steps to synthesizing a Parsel program, which we explicate in this section. We visualize the details in Fig.1 and provide a high-level pseudocode in Fig. A.11.

With Parsel, we balance two challenges. On one hand, function implementations must be modular to fully leverage combinatoriality. By requiring function signatures and descriptions to be specified (or generated), we ensure functions can call others without specific implementation information. On the other hand, it must be possible to avoid combinatorial explosions. By factoring the dependency graph into strongly-connected components (SCCs), we ensure the complexity grows exponentially only with the size of the largest strongly-connected component but not the total number of functions.

### 2.1 Specifying Parsel Programs

To formally specify a high-level algorithmic design, we develop a simple intermediate language, also called Parsel. It is designed to be accessible to programmers, code LLMs, and students, as discussed further in Appendix A, and inspired by many works (Section 4). In Parsel, each line contains a function description, a constraint, or a reference to a description. They all obey scoping rules, with some nuances per target language. In short: **descriptions** specify what a function does in natural language, optionally preceded by a function name and its arguments (if not included, they are inferred); **references** indicate to the language model that a function should call another function described elsewhere; **constraints** validate that a function is implemented correctly (if not provided, they may be generated automatically using the language model). We elaborate on syntactic details in Appendix R. We provide some examples of Parsel programs in Figures 2 and 3. For example, in Figure 2, `task_plan(): return a list of strings ...` represents a description, while the call to `collatz_recursion` by `recursion_rule` is a reference.

```
1 task_plan(): return a list of strings that
   ↪ represents an action plan to put a mug on
   ↪ the stall and bread on the desk.
2 -> "executable"
3   put_object_on(object, place): return a list
      ↪  of strings that represents an action
      ↪  plan to put an object in a place.
4   "mug", "stall" -> "executable"
```

```
1 # return a list of strings that
   ↪ represents an action plan to
   ↪ put a mug on the stall and
   ↪ bread on the desk.
2 def task_plan():
3    return put_object_on("mug", "
      ↪ stall") + put_object_on("
      ↪ bread", "desk")
4 # return a list of strings that
   ↪ represents an action plan to
   ↪ put an object in a place.
5 def put_object_on(object, place):
6    return [
7       'find ' + object,
8       'grab ' + object,
9       'walk to ' + place,
10      'put ' + object + ' on ' +
         ↪ place
11   ]

12 from execute_virtual_home import
    ↪ test_script; assert
    ↪ test_script(task_plan()) == "
    ↪ executable"
13 from execute_virtual_home import
    ↪ test_script; assert
    ↪ test_script(put_object_on("mug
    ↪ ", "stall")) == "executable"
```

Figure i: Parsel to VirtualHome (robotic planning)

```
1 collatz_recursion(num, cur_list=list()): Calls
   ↪ base_case if 1, otherwise recursion_rule
2 19 -> [19, 58, 29, 88, 44, 22, 11, 34, 17, 52,
   ↪ 26, 13, 40, 20, 10, 5, 16, 8, 4, 2, 1]
3   base_case(num, cur_list): Returns the list
      ↪ with the number appended to it
4   recursion_rule(num, cur_list): Add num to
      ↪ list, collatz with 3n + 1 if odd or n
      ↪ / 2 if even
5      collatz_recursion
```

```
1 # Calls base_case if 1, otherwise
   ↪ recursion_rule
2 def collatz_recursion(num, cur_list
   ↪ =list()):
3    if num == 1:
4       return base_case(num,
         ↪ cur_list)
5    else:
6       return recursion_rule(num,
         ↪ cur_list)
7 # Returns the list with the number
   ↪ appended to it
8 def base_case(num, cur_list):
9    cur_list.append(num)
10   return cur_list

11 # Add num to list, collatz with 3n +
    ↪ 1 if odd or n / 2 if even
12 def recursion_rule(num, cur_list):
13    cur_list.append(num)
14    if num % 2 == 0:
15       return collatz_recursion(num
         ↪ / 2, cur_list)
16    else:
17       return collatz_recursion((3
         ↪ * num) + 1, cur_list)
18 assert collatz_recursion(19) == [19,
    ↪ 58, 29, 88, 44, 22, 11, 34,
    ↪ 17, 52, 26, 13, 40, 20, 10, 5,
    ↪ 16, 8, 4, 2, 1]
```

Figure ii: Parsel to Python

Figure 2: Examples using Parsel to compile programs for VirtualHome [50] procedures and Python. Note the columns on the generated examples are only there to allow them to fit compactly – each program implementation is one contiguous solution. Colors for Parsel code on the left are used to indicate constraints and references. All other Parsel lines shown are definitions.

## 2.2 Generating Parsel Programs

We use a language model to generate Parsel programs from arbitrary natural language task descriptions as a few-shot translation task (or zero-shot with GPT4 [44]). Explicitly writing out a preliminary plan can be helpful, so we first ask the model to (zero-shot) generate a plan in natural language by thinking step-by-step (motivated by Kojima et al. [32]) and then prompt the model to translate the plan to Parsel. These steps correspond to the first and second arrows in Figure 3, respectively. In some domains, in particular for robotic planning, we find that this intermediate step is not necessary (while increasing cost) and can ask the language model to directly translate the task description to a Parsel solution.

## 2.3 Implementing Parsel Programs

Given the Parsel program, a core technical challenge is how to implement it in a modular way (to fully leverage the advantage of decomposition). Here we propose the Parsel synthesizer: At a high level, to implement Parsel programs with the Parsel synthesizer, we first use a language model to generate a set of candidate implementations of each of the functions based on their descriptions and then search over minimal sets of combinations of the functions to find ones that satisfy the provided constraints. The step described in this section corresponds to the rightmost arrow in Figure 3.

### 2.3.1 Implementing Functions

A function is implemented by first aggregating descriptions and function signatures of its (potentially zero) children to form an LLM prompt (for Python, we generate a prompt as if the child functions are imported and use their descriptions as comments). Crucially, this facilitates easily changing child implementations. A code LLM is then queried using the description's text as a docstring and the description's function name and arguments for the signature; full prompts shown in Appendix K.

### 2.3.2 Searching Over Function Combinations

**Sequential case.** The most straightforward case is when all functions have constraints (e.g., unit tests), and no functions have recursive dependencies (e.g., Fig. A.9). We start by considering this case. This defines a clear topological order on functions so they can be implemented sequentially. In this situation, Parsel implements functions with post-order traversal from function at the root, generating implementations and finding one passing the specified constraints for each function. In

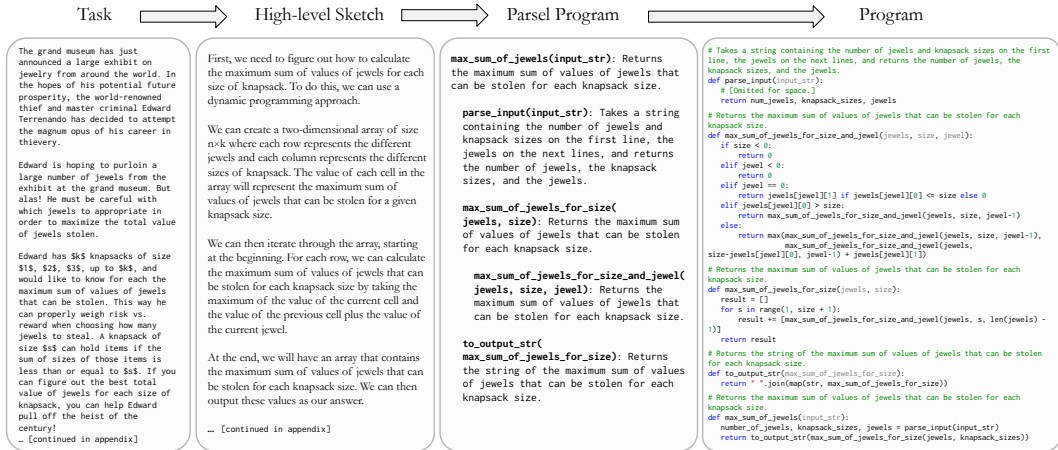

Figure 3: **Parsel pipeline.** Overview of the full Parsel pipeline for an APPS example. Each step is performed by a language model. We include the complete task, sketch, and program in Appendix S.

other words, without any cycles in the call graph, we can start by implementing the leaf functions first, then their parents, etc., until the program is implemented. However, in practice, many programs have more complex structures and constraints.

**Recursion.** Permitting mutual recursion (e.g. a function $f$ that depends on $g$ while $g$ also calls $f$) introduces the need for joint implementation of functions. However, naively considering all possible implementations is intractable for large programs. Specifically, the number of possible implementations of a program with $k$ functions and $n$ implementations per function is $O(n^k)$, exponential in the number of functions. We propose a more efficient solution, inspired by Cocke and Kennedy [16]. We reference the function call graph, identify its strongly-connected components (SCCs), and for each SCC consider all possible sets of function implementations until one set satisfies all constraints. For example, for possible implementations of functions $f, g, h$ forming an SCC of a call graph, with $I(f)$ corresponding to the language-model-generated implementations of $f$, we evaluate uniformly random samples without replacement from $I(f) \times I(g) \times I(h)$.

**Functions without constraints.** For functions with no constraints, we can conveniently use the same approach as above by reformulating the call graph as a "test dependency graph" and adding an edge from a test-less child to its parent. That is, if a function has no constraints, it depends on its parents to enforce constraints on its implementations. This could also allow us to automatically introduce new children via automatic decomposition without needing to also generate constraints for those children (see Subsection R.3). Alternatively, we support automatic test generation, prompting the language model to generate tests. For generated tests, we select the best implementation based on CodeT score [11], as some generated tests could be incorrect. In practice, we use the first (dependency) approach for inner functions and the second (test generation) approach when the top-level function has no tests. Note an opposite extreme from the sequential case: if no function or only the top-level function has constraints, we simply implement each function some number of times and then test all combinations. More generally, if $n$ functions with $m$ implementations where the sizes of the SCCs are at most $c$, partitioning reduces necessary evaluations from $O(n^m)$ to $O(\frac{n^c m}{c})$.

## 3 Experiments

We explore the ability of Parsel to generate programs for various tasks, including competitive coding, a standard coding benchmark, and robotic task planning. These three categories represent related but distinct kinds of algorithmic reasoning, with varying levels of abstractness and generalizability. Competitive programming represents one of the few benchmarks for evaluating code generation tools that benefits greatly from decomposition.[1] By evaluating Parsel on this breadth of tasks, we hope to better understand its generality as a framework for algorithmic reasoning.

---

[1]Unfortunately, benchmarks that evaluate language models on complex real-world programming tasks requiring thousands of lines of code do not currently exist. Developing such a dataset is an important direction for future work.

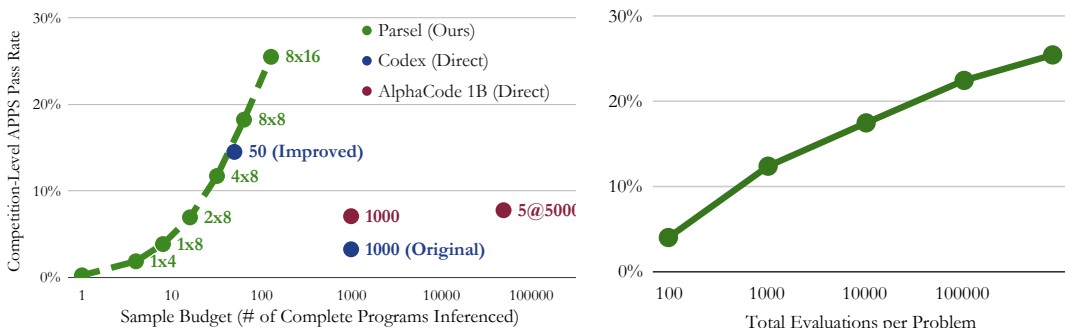

Figure 4: **Left: Competition-level problem pass rate**. Comparison of Parsel's pass rate on competition-level APPS problems [27] against direct code generation with Codex [12, 11] and AlphaCode [35] Labels correspond to pass@$n$ and pass@$n \times k$. Sample budget is measured in effective number of complete programs generated by the code LLM. (These measures are further explained in Subsection 3.1.) **Right: Pass rate vs number of evaluations.** Parsel generates and evaluates many programs with a small inference budget, by combinatorial composition. Though evaluation is cheap compared to generation, we examine to understand the effect of evaluation number. For evaluation budget analysis, we evaluate pass@$8 \times 16$ on a random 200-problem competition-level APPS subset, and sample combinations of function implementations at random for a given budget.

## 3.1 Python Code Generation

**Solving competition-level problems.** We evaluated Parsel on the competition-level subset of the APPS [27] dataset as follows: first, we zero-shot prompt GPT-3 (text-davinci-002) with the problem statement, requiring it to first propose a high-level solution and explain why it is correct, asking it to "think step by step to come up with a clever algorithm" [32]. We then prompt Codex [12] to translate the generated solution into Parsel code, providing three examples of valid code. We then attempt to synthesize the Parsel code and evaluate implementations to measure their pass rate, i.e., the proportion of solved problems. We report "pass@$n \times k$", where $n$ is the number of Parsel (i.e. intermediate, semi-formal) programs generated and $k$ is the number of implementations per function. Prior work has emphasized performance as a function of the LLM sample/inference budget measured in complete programs generated [12, 35]; for Parsel, we use the effective number of complete programs generated from the LLM for comparison, which is $n \times k$. We compare to directly generating solutions from Codex [12, 11] and AlphaCode [35]. We visualize the pass rate in Figure 4, finding that Parsel substantially improves over prior work [11], from 14.5% to 25.5%. We include full prompts and further evaluations in Appendix N and discuss the evaluation runtime costs in depth in Appendix T.8.

Since Parsel will "mix-and-match" function implementations, the number of distinct candidate programs available to test grows exponentially with $k$, while the generation cost for the components is linear. Generating each program likely requires on the order of petaFLOPs of compute [21], dwarfing the computational cost of testing complete programs. Yet when few functions have constraints (as in this evaluation) or there are complex recursive dependencies, it could become expensive to exhaustively evaluate all possible programs. Thus, we also explore the performance of Parsel as a function of the number of evaluations, as shown in Fig. 4. We find that the performance improves log-linearly with the number of evaluations, justifying spending compute to evaluate all programs implied by generated function implementations. One can interpret this to suggest the number of Parsel programs and combined implementations play similar and important roles. Thus, one may compensate for implementation ability with Parsel-generation ability and vice-versa (to a limit).

**Ablating the Parsel synthesizer.** We perform an ablation to explore the impact of the Parsel intermediate language: we provide the same high-level plans to Codex, but instead of translating them to Parsel and synthesizing programs as decribed, we translate them to Python directly. We match the code generation budget (effectively complete programs inferenced), generating 16 Python implementations[2] per high-level plan on 100 randomly sampled problems, and find that the pass@$8 \times 16$

---

[2]Sampling the longer solutions resulted hitting a rate limit more frequently, even when using the same number of total tokens per prompt; a full 128-solution comparison on 1000 problems would potentially take months.

performance drops to 6% from 25.5% for the full pipeline. Not only did this approach solve substantially fewer problems, but the problems that it did solve were a strict subset of those which we were able to solve with Parsel. This indicates that for Parsel with Codex, the step-by-step decomposition provided by the high-level plan is not by itself responsible for the observed improvements by Parsel.

We additionally perform an ablation to understand how Parsel's performance without a high-level plan. On a 200-problem competition-level APPS sample, we found that, without a high-level plan, the accuracy falls to 13% (from 25.5%, with the same configuration). This is better than the ablation where Parsel wasn't used, suggesting that Parsel plays a larger role than the high-level plan on APPS, but both are necessary. Note part of this improvement is from the challenge of prompting Codex to generate Parsel, given long APPS problem statements.

**HumanEval.** We next tested Parsel on HumanEval[12]. Because these problems are substantially simpler, they provide an opportunity to evaluate pass@1 by generating tests automatically. Furthermore, this smaller and simpler set of problems enables us to assess the (much more costly) GPT-4. We found a significant improvement over the state-of-the-art pass@1 performance—from 67% to 85%. We found that by describing the Parsel format (see Appendix O for details), noting the role of indentation in providing hierarchy and explaining the format of a description, GPT-4 was able to translate natural language plans into Parsel zero-shot. Moreover, the combination of Parsel and GPT-4 surpassed GPT-4 alone on HumanEval, both with and without generated tests, further emphasizing the importance of leveraging intermediate languages like Parsel for improved code generation performance in complex tasks.

The 85.1% pass rate with generated tests, evaluated based on the CodeT score [11], was obtained by sampling until three Parsel programs had one implementation passing a generated test and then selecting the one passing the most tests. On average, this required 3.5 Parsel programs per problem, with 8 implementations each (i.e. < 32 complete implementations on average). Parsel also outperforms GPT-4 with CodeT alone on the same set of tests, and using a comparable 32 samples, which passes 81.1% of the problems.

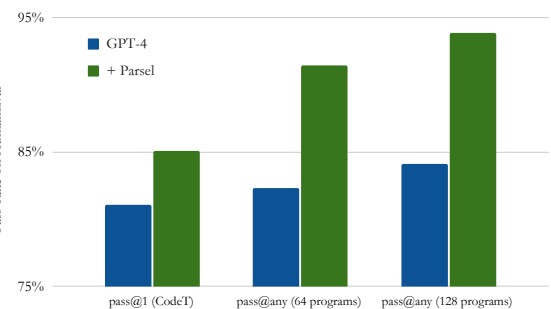

Additionally, when evaluating performance in terms of pass@any (i.e., not filtering attempts by CodeT), we discovered that by allowing up to 8 Parsel programs (still with 8 implementations each), the pass rate increased to 91.5%. In contrast, when allowing up to 64 directly generated programs, the pass rate was only

Figure 5: **Pass rates on HumanEval.** We analyze the effect of using Parsel with GPT-4 [44] on HumanEval [12]. Given the same program inference budget, the pass rate with Parsel improves regardless of whether one filters by generated tests.

82.3%. This difference becomes more pronounced when generating 128 programs (128 directly vs. $16 \times 8$ with Parsel), improving from 84.1% to 93.9% (See Fig. 5).

While there are still eight tasks that are never solved with or without Parsel, GPT-4 solves eighteen problems with Parsel that it cannot solve alone. For eight of these, the solution has six or more functions (excluding defined but unused functions). Of the problems Parsel solved, the ones GPT-4 could not solve itself required 4.2 functions on average, while those GPT-4 could solve required only 2.8 functions (sampled over 50 problems). We find GPT-4 alone solves two problems pass@any that GPT-4 with Parsel cannot: `decimal_to_binary` and `decode_cyclic`. The `decode_cyclic` solutions match Chen et al. [12]'s Figure 2 (but not the dataset's canonical solution) with minor comment differences, indicating possible contamination. In `decimal_to_binary`, the generated Parsel solutions often unsuccessfully rely on statefulness: for example, `divide_by_2_with_remainder` calls `calculate_remainder` and `update_decimal`, but `update_decimal` cannot mutate an integer.

Most crucially, these experiments indicate that, even when given the same set of generated tests and program generation budget and selecting only one best solution, Parsel significantly increases the probability that the solution is correct on the ground-truth set of tests.

**Comparison to a human expert.** In our fully automated pipeline, an LLM generates a Parsel program, which Parsel attempts to synthesize into a solution. We also evaluated how well an expert Parsel user would perform by directly writing Parsel and interacting with the Parsel synthesizer. As a case study, one of our authors with significant prior experience in competitive programming was presented with 10 randomly-selected competition-level Codeforces problems from the APPS dataset, writing Parsel from scratch in order to solve the problems The participant successfully solved 5 of the problems within a 6-hour time frame.

In comparison, when GPT-3 generated Parsel solutions to these problems, it had a success rate of 2 out of 10 problems with 8 attempts per problem (of course, in a much shorter time-frame). This result suggests that, given a suitable decomposition, the Parsel synthesizer can effectively generate complete correct solutions to hard problems – thus, a major point for improvement in the fully automated pipeline lies in the first stage (Parsel generation). In other words, improving the ability of models to decompose tasks into Parsel programs is a key bottleneck and is an important direction for future work.

While the participant could solve problems that GPT-3 could not, the participant found some aspects of working with Parsel counterintuitive. For example, one of the most effective ways to provide additional details to ensure a working solution was to write additional tests rather than to clarify the meaning in the function descriptions. Given the stochasticity of language model generation, there was concern that changing the description of a child function could break its parent – however, in practice, this seemed rare. We include the participant's solutions and the associated problems in Appendix Q.

**Chained components.** Chen et al. [12] highlight in their discussion of limitations that language models fail to produce code that requires the simple composition of building blocks. We replicate their experiment with Parsel, using the same docstring and providing the same descriptions as function descriptions. In Fig. 6, we visualize how Parsel solves this. We find that as the number of required components grows, even interchanging the components of only two complete programs (1x2) solves more problems than Codex solves with 16 programs. This illustrates a key point: assuming any independence, the complete set of combinatorial combinations of $n$ implementations of a program's functions is clearly more likely to be correct than $n$ samples from that same set. Moreover, this suggests that a framework like Parsel may support increasingly complex and abstracted programs as language models improve.

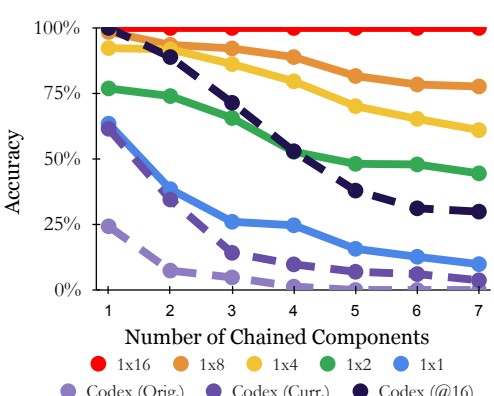

Figure 6: **Pass rate vs number of chained components.** Parsel performance is shown as 1x$k$ where $k$ is the number of complete programs sampled. Orig and Curr are the original [12] and current Codex results, respectively. Codex (@16) corresponds to the pass rate with 16 programs.

### 3.2 Robotic Planning in VirtualHome

We perform a study on VirtualHome [50] to demonstrate that Parsel can also be used for complex robotic planning. VirtualHome is a simulation environment consisting of households with various interactive objects and agents. VirtualHome includes a small set of permissible actions with a strict grammar – tasks like "paint ceiling" may require multiple levels of decomposition, e.g., finding and collecting specific painting supplies, finding and using a ladder, and using the painting supplies, each requiring further decomposition.

To test the effectiveness of Parsel in this domain, we investigate whether Parsel could generate programs to solve tasks in the VirtualHome environment, while using the environment to provide feedback on whether the plan is executable. Specifically, we use Parsel to generate a Python program that can generate action plans in natural language similar to ones used in Huang et al. [28]. In each specified constraint, the produced natural language action plan is translated to formal VirtualHome instructions with minimal regex matching and tested executability. If the instructions can be successfully executed, they are considered valid – however, a potential enhancement could be the inclusion of object-relational constraints on the subsequent state of the world. We include an example of a Parsel program that successfully executed and decomposed a task in VirtualHome in Figure 2. Note we also used a header describing a valid action plan, shown in Figure A.45.

However, as pointed out by [28], while it is easy to check plan executability, it is harder to check correctness, because there are generally many valid ways of accomplishing realistic tasks. Thus, like Huang et al. [28], in order to evaluate the accuracy of the executable plans, we perform a human study. Specifically, we ran two surveys on Prolific [45] where we asked 20 participants to make five rankings each, for a total of 100 rankings per survey. In one survey, we asked participants to rank a set of solutions by how accurately each solution accomplishes the task while in another we asked about how understandable the solution was. Two versions of distinct Parsel solutions were shown (where multiple executable Parsel solutions were available), one which included the indented function names used to solve the tasks alongside their step-by-step solutions, and one which only included step-by-step output. We compared both Parsel solutions to an also-executable baseline solution, based on Huang et al. [28]. We include more details about the survey format and executability in Appendix P.

Humans ranked the Parsel solutions as more accurate than the baseline. In each ranking, both the indented and non-indented Parsel solutions were consistently ranked higher than the baseline. In accuracy, standard Parsel solutions (as well as indented solutions) were identified as more accurate than the baseline solutions in 69% of comparisons, more than twice as often as the baseline. In clarity, the standard Parsel solution was 70% more likely to be preferred over the baseline while the indented Parsel solution was 50% more likely to be preferred compared to the baseline. There was no notable difference between indented and non-indented Parsel solutions in either accuracy or clarity. Specifically, note that all of these comparisons are reported pairwise, and Figure 7 compares the rankings given to the (non-hierarchical) Parsel-generated plan and the Codex-generated solution.

In essence, participants did not appear to prefer either Parsel plan format over the other, but given a choice between one Parsel-generated plan and its corresponding Codex-generated plan, they typically considered the Parsel-generated plan to be more accurate and clearer.

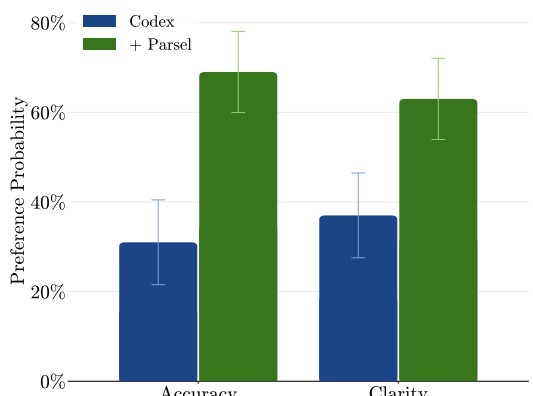

Figure 7: **Robotic plan comparison.** Parsel-generated robotic plans were consistently selected as more accurate and clear when compared to a direct generation approach like in Huang et al. [28].

## 4  Related Works

**Step-by-step problem solving with LLMs.** Many works show that step-by-step reasoning benefits LLM performance [51, 54, 42, 60, 37, 34] and correspondingly, that this performance can be improved with guidance and tool use [71, 68, 63, 59, 22]. Acquaviva et al. [2] encouragingly showed that humans, when asked to explain how to solve problems in the Abstract Reasoning Corpus [14], tended to provide step-by-step hierarchical descriptions with many verification steps. Moreover, Wies et al. [61] presents a theoretical argument showing problems that can be learned efficiently if decomposed but require exponentially many examples w.r.t. length if not decomposed.

**Program synthesis.** Program synthesis is the long-standing challenge of generating programs from high-level specifications [26], such as input-output examples [7, 25] and/or natural language descriptions [52, 65, 19]. Program synthesizers typically search the exponentially large space of programs. Consequently, synthesizing large, complex programs remains an open challenge. Recently, *library learning* has shown a way to make progress: even complex programs can be short in terms of the right high-level library. In turn, this library can be progressively induced from solutions to simpler synthesis problems. This idea is embodied in DreamCoder [23, 9]. Library learning requires a rich distribution of related tasks so that patterns emerge from solutions to simple problems. Patterns are abstracted into useful library functions, enabling short solutions to more complex problems. Parsel also aims to synthesize complex programs by decomposing them. However, Parsel programs specify decompositions, so a family of related tasks is not required.

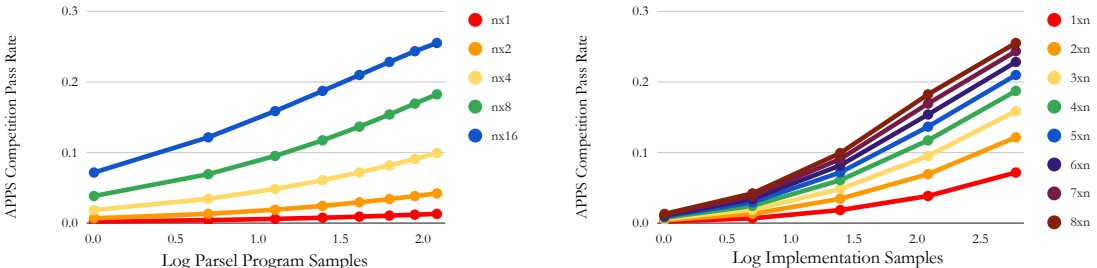

Figure 8: **Scaling performance.** The probability of passing an APPS competition-level problem increases quadratically with respect to the log of the number of Parsel programs sampled (left) and the number of implementations sampled (right).

**LLMs for formal environment multi-step planning.** Also encouragingly, several existing works can be expressed in Parsel. For example, Huang et al. [28] and Brohan et al. [10] showed that language models can automatically generate step-by-step algorithms for robotic agents in language. In both cases, the generated language corresponds directly to pre-implemented low-level robotic abilities. This could be expressed by providing a task description and constraints evaluating that the high-level task was completed successfully. In addition, Jiang et al. [30] proposed a framework to generate formal proofs in formal theorem-proving languages from informal proofs by first generating an intermediate natural language proof sketch. This could be expressed in Parsel by generating each sketch step as a function and using formal verification for each lemma as the Parsel validation step.

**Programming languages and frameworks incorporating language models.** Other works have explored programming languages that incorporate language models. For example, Cheng et al. [13] explored the introduction of a language-model-based evaluation function, which would allow `f('North America?', 'U.S.')` to automatically return 'yes' by referencing the knowledge of the language model, and showed that they could also generate programs using this tool with a language model and Beurer-Kellner et al. [8] explored a related SQL-style LLM-querying language. In addition, Dohan et al. [20] presents an inference-focused framework for language models more broadly for probabilistic graphical models composed of language model actions. Unlike LM Cascades [20], we primarily focus on the constrained generation of programs, instead of leveraging language models as functions within a particular program.

**Testing code language model outputs.** Related works have explored the capacity of assert statements to constrain the generation space of code LLMs on functions [6, 12, 35]. The automatic generation of unit tests in Chen et al. [11] allowed them to filter many programs before performing final evaluations, significantly improving the pass rate per evaluation. CodeT [11] applies a language model to generate a large number of tests, presuming that some of them will be incorrect. However, instead of simply selecting the program passing the most tests as correct, CodeT groups together generated programs by the subset of the tests that they pass – it then returns any program from the group with the most tests passed in total [11]. Chen et al. [11] intuitively motivates this by the Anna Karenina principle: "Happy families are all alike; every unhappy family is unhappy in its own way."

Notably, both the number of final evaluations and the sample budget are important, and we find that combining Parsel with automatic test generation improves performance given a fixed sample and evaluation budget. In addition, Merrill et al. [38] proves essential constraints on what can be learned from assertions alone and, more crucially, what cannot.

## 5   Discussion and Limitations

We note that Parsel has important limitations, largely stemming from its dependence on closed LLMs. First, LLMs naturally underperform on languages underrepresented in their training data, like Parsel, affecting the quality and reliability of the generated code, and these closed models do not offer a training API. In addition, reliance on Codex and GPT-4 introduces vulnerabilities due to potential abrupt changes in behavior – since this project's start, access to Codex (which is free) has become much more limited and the much-better GPT-4 is also much more expensive. We unfortunately found a 2.7B CodeGen model [39], evaluated like Codex on APPS, solved no problems. However, we anticipate that improvements in open-source code LLMs will mitigate these issues.

Despite the limitations, elaborated in Appendix B, and the challenges of using a language model to generate a language that it has never seen before, the generated Parsel was able to implement robust algorithmic reasoning. There are a few natural questions that arise from this work. How robust is Parsel to the language used to specify the problem? When solving hard problems, is it better to increase the number of Parsel programs sampled or the number of program implementations sampled? We visualize Parsel's performance on the explored subset of APPS in Figure 8 to try to investigate some of these questions. In general, it appears that the probability of passing increased quadratically ($R^2 \geq 0.99$ for all curves) with respect to the log of both the number of Parsel programs sampled and the number of implementations sampled. Clearly, since the pass rate is bounded between 0 and 1, this trend cannot continue indefinitely. However, given the rate limit associated with calls to the Codex API (officially, 40,000 tokens or 20 requests per minute, but in practice, we consistently ran into the rate limit with $\approx 10\%$ of that usage), we were not able to identify the inflection point. Li et al. [35] indicated the pass rate curves become log-linear, but we cannot guarantee this holds for Parsel.

Furthermore, we highlight that the question we (and, in our view, most prior work) were looking at in our APPS experiments is, given hard problems and limited by one's generation budget, what is the chance of finding a correct solution? However, if one instead views each test as a successive submission, where any failure is comparable to a crash in production, one would likely take an approach like the one in our HumanEval test generation experiments, one that uses interpreter feedback to revise solutions, or perhaps even one that uses the language model to simulate the result of tests. Both interpretations are important, but they highlight different concerns: even if one can guarantee success upon submission given a trillion generated tokens, this has limited practical applicability. A method's ability to efficiently generate reasonable solutions in the first place is one key bottleneck, and assuming an imperfect model, the ability to identify them is another.

Finally, we highlight that Parsel is already capable of synthesizing larger, more complex programs. For reference, we include a working Lisp interpreter in Appendix I, based on Norvig [41], which we translated to Parsel with 21 functions. We make sure to include no explicit references to Lisp in the Parsel code and provide a header to tell Parsel to keep track of environments as dictionaries. However, this is clearly an imperfect substitute for a true object-oriented programming implementation, and we believe adding better object-oriented features to Parsel would be a valuable future direction.

# 6 Conclusion

We hope that Parsel provides a broadly useful framework for several groups: for programmers, Parsel should provide a language for robust code generation without the need to evaluate the underlying code; for students, Parsel should allow the teaching of algorithmic reasoning with less emphasis on syntax and more emphasis on problem-solving, similarly to a mathematics curriculum; for language models, Parsel should facilitate hierarchical task decomposition.

Ultimately, as we discuss in detail in Appendix C, Parsel has many natural extensions building on prior work in code synthesis from automatic test case generation [17, 12], reranking solutions by model-judged quality [69], more clever recursive decomposition [49], bootstrapping increasingly complex programs [4, 43, 68], to generating language model prompts as part of a target language [20, 31]. As we discuss in Appendix D, we also envision that Parsel may be useful for theorem proving. Thus, Parsel helps fill the gap between reasoning and execution, providing a new approach to complex, hierarchical reasoning tasks for language models.

# Acknowledgements

We would like to thank Jesse Mu, Tianyi Zhang, Rose E. Wang, Allen Nie, Ben Prystawski, Xindi Wu, Fan-Yun Sun, Isaac Kauvar, Xi Jia Zhou, John Thickstun, and Shikhar Murty for their helpful feedback, as well as Rishi Bommasani for highlighting highly relevant works. We also thank the Stanford HAI-Google collaboration for their Cloud Credit grant. Qian Huang is supported by an Open Philanthropy AI fellowship. Gabriel Poesia is supported by a Stanford Interdisciplinary Graduate Fellowship. In addition, this work was partially supported by National Science Foundation Grant No. 2302701 and National Science Foundation Expeditions Grant No. 1918771.

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

```
1 select_airport_cities(city_road_cost, city_airport_cost): given a matrix representing the
  ↪ cost of building a road between any two cities, and a list representing the cost of
  ↪ building an airport in a city (where any two cities with airports are connected), return
  ↪  a list of the cities that should have airports built in them to minimize the total cost
  ↪  of building roads and airports such that all cities are connected. The list should be
  ↪ sorted in ascending order.
2 [[0,3,3],[3,0,3],[3,3,0]],[0,0,0] -> [0,1,2]
3 [[0,3,3],[3,0,3],[3,3,0]],[10,10,10] -> []
4 [[0,10,3],[10,0,11],[3,11,0]],[1,4,5] -> [0,1]
5     sky_city_cost(city_road_cost, city_airport_cost): given a list of lists representing the
      ↪ cost of building a road between any two cities, and a list representing the cost of
      ↪ building an airport in a city, return a new cost matrix with a new node corresponding
      ↪  to the sky.
6     [[1,2,3],[1,2,3],[1,2,3]],[4,5,6] -> [[1,2,3,4],[1,2,3,5],[1,2,3,6],[4,5,6,0]]
7     minimum_spanning_tree(cost_matrix): given a list of lists representing the cost of each
      ↪ edge, return an adjacency matrix corresponding to the minimum spanning true.
8     [[0,1,3,4],[1,0,2,100],[3,2,0,5],[4,100,5,0]] ->
      ↪ [[0,1,0,1],[1,0,1,0],[0,1,0,0],[1,0,0,0]]
9     final_node_connectors(adjacency_matrix): given a list of lists representing an adjacency
      ↪ matrix, return a list of the nodes connected to the final node. However, if only one
      ↪ node is connected to the final node, return an empty list.
10    [[0,1,0,1],[1,0,1,0],[0,1,0,0],[1,0,0,0]] -> []
11    [[0,1,0,1],[1,0,1,0],[0,1,0,1],[1,0,1,0]] -> [0,2]
```

Figure A.9: A potential programming assignment focused on problem-solving rather than implementation. The top-level function and asserts would be the assigned problem (which Codex [12] does not seem to be able to solve directly), while the other functions would be the student solution.

# A   Implications

Parsel is a natural language compiler framework that bridges the gap between natural language and programming language by allowing programmers to write high-level algorithmic designs in natural language and automatically compiling them into valid code. This has potential benefits for programmers, students, and code language models.

## A.1   For Programmers

### A.1.1   Current Limitations

First, programming generation language models like Codex continue to be constrained primarily to individual functions, rarely exceeding a few dozen lines in practice [12, 58]. This is still a dramatic shift from foundational earlier works, which focused on the association between one line of natural language pseudocode with one line of code [33] or a line of text to a StackOverflow snippet [66]. Yet, these models perform worse the more unusual the desired functions are, and recent research suggests that people using these language models are more likely to introduce buggy code [47], although this is not yet conclusive [53].

### A.1.2   Potential Benefits

On the other hand, results from Google and others indicate that professionals can write code more efficiently with large language models, and the benefits will likely only improve as they improve [58]. Since Parsel requires constraints that ensure functions behave as expected, this should encourage bug-free programs and avoid the need for manually checking that specific underlying functions are correct. Furthermore, a function written in Parsel is likely to be more resilient to breaking changes in the target language, especially syntactic changes (e.g. Python2 to Python3). In addition, a natural extension would draw on work on automatic unit testing [17] to suggest additional constraints where behavior is ambiguous between implementations of a function. However, for Parsel to be practically useful for real-world programmers, we expect multiple improvements to be necessary, including additional object-oriented features.

### A.2 For Students

#### A.2.1 Current Limitations

In addition, these language models pose serious challenges for programming pedagogy – existing introductory programming classes rely extensively on teaching syntax and how to implement algorithms over how to solve problems with them. Free language model-based tools like Copilot can essentially solve many of these introductory assignments directly, function by function. Those which cannot be solved currently will be increasingly solved [18].

#### A.2.2 Potential Benefits

Many students currently introduced to programming struggle with learning syntax and debugging unclear compiler or interpreter errors. However, abstracting away these details with a natural-language coding language will likely make learning to code more accessible to students who are just beginning to code. In addition, stepping away from implementation-focused assignments will allow a focus on higher-level problem-solving assignments earlier. These will allow for assignments that are more like those in mathematics. For example, for a problem like Figure A.9, instead of choosing between requiring students to manually implement a problem-solving focused question like the top-level description of, or requiring teaching assistants to manually evaluate the reasoning for correctness, one could ask them to implement a solution in Parsel.

### A.3 For Code Language Models

#### A.3.1 Current Limitations

Traditional programming languages result in some unique challenges for language models. For example, unlike natural languages, traditional programming languages are far less robust to slight variations in wording. In addition, traditional programming languages require many tokens for syntactic details and in some cases, may take many lines to express what can be expressed far more simply in language. For example, referring to a shortest-path algorithm or Conway's game of life takes far fewer tokens than actually implementing them. However, even with fairly nonstandard problems, LLMs have shown remarkable algorithmic generalization ability [36, 62, 3, 72]. One alternative that has been explored is conversational code generation [40, 67]. However, these approaches have primarily focused on highly imperative programming structures. Moreover, they still require having the full program in context and do not clearly generalize to complex hierarchical programs with many functions.

#### A.3.2 Potential Benefits

Parsel allows code language models to stay closer to natural language when generating code, which corresponds more closely to their primary source of training data. Moreover, it allows complex but standard methods to be described concisely, requiring fewer tokens to generate. One exciting additional benefit is the potential to generate solutions recursively: if the Parsel compiler is unable to find a solution for a set of functions, it should be possible to prompt the model to define new helper functions. In fact, we find that often the model attempts to reference undefined auxiliary functions when defining complex functions (e.g. "count_living_neighbors(grid, i, j)" in Conway's game of life), and as a result support an optional argument where the model can attempt to resolve NameErrors automatically by attempting to implement functions.

## B   Limitations

There are several limitations to the current implementation of Parsel. First, Parsel relies on a code LLM to generate implementations of individual functions, and the quality of these implementations can vary depending on the specific model used and the complexity of the function descriptions. In particular, Parsel may struggle to generate correct code for individual functions with complex behavior (i.e. functions that Codex cannot implement). However, this can be mitigated by decomposing the complex functions into simpler ones that can be implemented more easily.

The current implementation of Parsel may struggle to generate correct code when there are many functions with complex dependencies or without constraints. This is because the number of implementation combinations to consider grows exponentially with the size of the largest strongly connected components. As discussed, this can limit Parsel's performance on some programs. However, approaches like Chen et al. [11] may be able to mitigate this.

Code LLMs, unfortunately, do not perform well on languages underrepresented in their training data – with few examples to learn from, LLMs may struggle to generate correct code in these languages [5]. However, some LLMs can learn new languages in context, allowing them to generate code in languages not in their training data [5]. These limitations can impact the quality and reliability of the code generated with Parsel. In addition, because code LLMs have never been trained on Parsel, this harms their ability to generate it. While we could wait for Parsel to gain widespread adoption, it should also be possible to translate many existing codebases to Parsel. We include a proof-of-concept backtranslation/decompilation study in Appendix L.

In addition, the best open-source code LLMs currently available e.g. PolyCoder [62] substantially underperform Codex, while Codex is competitive with other traditional LLMs on reasoning tasks [36]. However, this dependence on closed models creates a vulnerability, as the providers of closed LLMs can change behavior (e.g. rate limits or model implementations) without warning. Indeed, between the time we started working on Parsel and this version of the paper, OpenAI ended widespread access to Codex, now available only by request.

Because of this, we evaluated a 2.7B CodeGen model from Nijkamp et al. [39] with Parsel in the same configuration we used when evaluating APPS on Codex (in the 8x16 configuration). We found that it could solve none of the random 25 problems which we evaluated it on. However, despite these limitations, the current Parsel implementation has shown promising results in generating correct code for a variety of functions and languages. Many limitations will likely be ameliorated as code LLMs improve.

## C Future Work

In the future, we hope to more deeply integrate automatic unit test generation, especially in combination with user-provided tests [17, 11]. One method would be to identify edge cases and check whether the set of functions that successfully solve all existing tests disagree on any new tests. This could permit automatic decomposition without exponential growth in implementation combinations. Techniques like those proposed in Zhang et al. [69], which would allow us to rerank a set of solutions, could also allow us to search the combinatorial space of solutions more quickly. Relatedly, for the robotic task planning, incorporating asserts at the execution level (e.g. checking whether the agent is close to the microwave, as in Singh et al. [56]) is a promising research direction. Furthermore, evaluating the examples in this paper, we found that using the minimum CodeT score across all generated functions was a consistently effective heuristic to identify good sets of functions. However, generating unit tests for all functions when generating Parsel programs instead of generating unit tests for a shared top-level function increases the inference cost from linear in the number of tasks to also being linear in the number of functions and Parsel programs generated. Finding a way to balance this tradeoff would likely be valuable.

In addition, we plan to incorporate ways of varying the "confidence threshold" of the language model. Ensuring that the descriptions are straightforward and unambiguous is important for more critical programs and parts of programs. In addition, when teaching students simpler concepts, requiring them to decompose the task further may be useful.

We would like to integrate value functions to allow decomposition to be done more methodically where no verification is possible. Specifically, automatically decomposing all functions that have not yet been implemented in an SCC is suboptimal and could be improved with a model of expected improvement due to expansion, as done for proof expansion in Polu and Sutskever [49]. In addition, when decomposing functions, we would like to permit the model to reference already-defined functions (rather than to just define new ones). We might even use the code language model to determine which function to evaluate next. Further, we aim to support more general reward functions for function implementations where multiple may be valid but we rank implementations based on a desired feature. These "soft" constraints may also allow new Parsel uses, e.g. planning stories in natural language [64].

```
1  and_commute(p q: Prop): the and operator is
         ↪ commutative
2  show (p ∧ q → q ∧ p) ∧ (q ∧ p → p ∧ q)
3    p_q_implies_q_p(p q: Prop): if p ∧ q, then q
         ↪ ∧ p
4    q_p_implies_p_q(p q: Prop): if q ∧ p, then p
         ↪ ∧ q
```

```
1  -- if p ∧ q, then q ∧ p
2  lemma p_q_implies_q_p(p q: Prop):
3
4     p ∧ q → q ∧ p :=
5  begin
6    intro h,
7    cases h with hp hq,
8    split,
9       exact hq,
10      exact hp,
11 end
12 -- Description: if p ∨ q, then q
         ↪ ∨ p
13 -- if q ∧ p, then p ∧ q
14 lemma q_p_implies_p_q(p q: Prop):
15
16    (q ∧ p) → (p ∧ q) :=
17 begin
18   intro h,
19   split,
20      exact h.right,
21      exact h.left,
22 end
```

```
23 /-
24   Theorem:
25     If q ∧ p, then p ∧ q
26 -/
27 -- the and operator is commutative
28 lemma and_commute(p q: Prop):
29   (p ∧ q → q ∧ p) ∧ (q ∧ p →
         ↪ p ∧ q) :=
30
31 begin
32   apply and.intro,
33   { apply p_q_implies_q_p },
34   { apply q_p_implies_p_q }
35 end
```

Figure A.10: Parsel to Lean (theorem proving)

Finally, we hope it would be possible to use Parsel as a framework for bootstrapping increasingly complex program generation (e.g. Anthony et al. [4], Zelikman et al. [68], Odena et al. [43]). That is, by 1) generating Parsel examples from a purely natural language specification and then reinforcing those which successfully compile, and 2) by reinforcing the model with each successfully compiled component, we would likely be able to iteratively improve performance with an arbitrarily large dataset of examples.

Another feature that would be valuable would be the ability to incorporate multiple base tools with different kinds of specialized models, inspired by Ibarz et al. [29] and Dohan et al. [20]. That is, it would be valuable to allow a model to determine which target language to use, possibly combining them. For example, for large parts of the Tensorflow and PyTorch libraries, while their interfaces are written in Python, they depend heavily on large C++ codebases [46, 1]. Relatedly, Cobbe et al. [15] showed that giving language models access to a calculator allowed them to solve more complex math word problems. This, combined with the observation that Parsel could also compile programs by generating language model prompts to be used as part of the program, may potentially allow the automatic generation of task-specific language model cascades [20].

Another noteworthy addition would be the integration of Synchromesh [48], ensuring that each new word or token generated by the model is actually possible within the grammar of the given formal language and does not violate other semantic constraints.

Ultimately, we hope that this specification for Parsel is a jumping-off point for a new way of thinking about programming and reasoning.

## D   Theorem Proving in Lean

With the same framework, we can generate proofs in formal theorem-proving languages such as Lean, as in Figure A.10. We include the translated version in the appendix. Note a nuance of Lean and theorem-proving languages is that the ability to run Lean on proof with no errors/warnings indicates the proof is correct (but is not a guarantee that the proof statement matches our claim in language). Thus, each function in a Lean Parsel proof has an "implicit constraint." This makes it straightforward to identify which informal parts of a proof are most difficult to explicate. Generally, we believe Parsel can be a powerful tool for theorem proving.

Yet, we observed important challenges in this context, which we believe are avenues for future work and can be resolved. For example, in datasets such as MiniF2F [70], many proofs require explicit calculations in intermediate steps. That is, many proofs are similar to "Find the minimum value of $\frac{9x^2 \sin^2 x + 4}{x \sin x}$ for $0 < x < \pi$. Show that it is 012." (from the informal MiniF2F introduced by Jiang et al. [30]). We believe that a dataset of proof statements (in natural and formal language), requiring complex proofs that are more abstract and less dependent on explicit calculations would allow us to better measure progress towards solving difficult theorems – we leave this to future work.

```
1  parsel(program, target_language): synthesize a program from a string specifying a Parsel program.
2    parse_program(program): parse the Parsel program string to a call graph
3      create_root_node(): create a root node as the current function node, without any constraints
4      parse_line(line, current_node, current_indent) -> function_graph: for each step up in indentation, set the current node
         ↪ to its parents. then, parse the definition, reference, or constraint.
5        parse_definition(line): create a new function node, make it a child of the current node's parent, then assign it as
           ↪ current node.
6        parse_reference(line): add reference as a child of current node if reference is an ancestor or a direct child of an
           ↪ ancestor
7        parse_constraint(line): add the constraint to the current node's constraints.
8    get_dependency_graph(function_graph) -> dependency_graph: taking the function graph, create a copy where all nodes without
       ↪ asserts also depend on their parents unless the target language implicitly tests all functions.
9    identify_strongly_connected_components(dependency_graph): return SCCs of the dependency graph and the edges between the
       ↪ SCCs.
10   synthesize_scc(scc, scc_graph): find an implementation string solving a given SCC, starting with SCC dependencies, then
       ↪ generating possible implementations of SCC functions, then finding an implementation combination satisfying the
       ↪ functions' constraints
11     synthesize_children(scc, scc_graph): synthesize any SCCs this SCC depends on and add them to the implementation string.
12       synthesize_scc
13     generate_implementations(scc, n, children_implementation_str): for each function in the SCC, prompt the language model to
         ↪ generate n implementations of each function starting with the implementation string of the SCC's children.
14     solve_constraints(scc, fn_implementations): taking the provided constraints of each function in the scc, evaluate a
         ↪ shuffled list of the direct product of implementations with the constraints until one passes all of them
15       direct_product_implementations(fn_implementations): return the direct product of the list of lists of
           ↪ fn_implementations
16       generate_constraints(fn_node): translate each of the constraints into an evaluation string idiomatic to the target
           ↪ language and add these to the list of combined implementations
17       eval_str(scc, implementation_str): evaluate an implementation including constraints by running it in a target-language
           ↪ executor
18     on_fail(scc, scc_graph): raise an error highlighting the scc which could not be synthesized
```

Figure A.11: Pseudocode in the style of Parsel describing how Parsel synthesizes programs. A detailed version including automatic decomposition and automatic infilling is in Figure A.12 of Appendix G. Constraints are left out for clarity – e.g. one could define a test function and validate the compilability (or lack thereof) of a set of reference Parsel programs.

# E    Optimizations

## E.1    Caching

We cache responses from the language model with respect to the prompt and language model decoding parameters 1) to reduce the number of queries necessary and 2) to keep the programs generated mostly stable (i.e. a working function should continue working unless it or its children change). To this end, when the number of desired implementations increases for a pre-existing query with all other arguments fixed (temperature, number of decoding tokens, etc), we append the additional ones to those already generated.

## E.2    Automatic Function Infilling

Sometimes, a function generated by a language model may call a function that is not yet implemented. In this case, we can (optionally) attempt to automatically generate and implement it based on its usage. The function is then incorporated into the call graph as a unit-test-less child of the function which calls it. To avoid infinite recursion and inefficient use of language model quota, we limit the number of times that this process can be applied to a function.

## E.3    Multiprocessing

We use multiprocessing with a user-specified timeout to test many implementation sets in parallel to allow for many fast solutions to be tested alongside slower solutions[3].

# F    Parsel Pseudocode

# G    Parsel Pseudocode

We include a longer-form Parsel pseudocode in the style of Parsel. Note this pseudocode does not include backtranslation.

---

[3]As anticipating the number of steps that a solution will take universally is a version of the halting problem and thus intractable.

```
1  parsel(program, target_language, allow_autofill=False, allow_autodecomp=False): compile a program from a string specifying a
   ↪ Parsel program.
2   parse_program(program): parse the Parsel program string to a call graph
3     create_root_node(): create a root node as the current function node, without any constraints
4     parse_line(line, current_node, current_indent) -> function_graph: for each step up in indentation, set the current node
      ↪ to its parents. then, parse the definition, reference, or constraint.
5       parse_definition(line): create a new function node, make it a child of the current node's parent, then assign it as
        ↪ current node.
6         parse_line_to_fn(line) -> name, args, rets, description: extract the function name, arguments, optionally returned
          ↪ variables, and description of the form "name(args) -> rets: description" if return variables are present else "
          ↪ name(args): description".
7         populate_fn_node(name, args, rets, description): populate the new node's name, arguments, description, and optionally
          ↪  a list of returned variables.
8       parse_reference(line): add reference as a child of current node if reference is an ancestor or a direct child of an
        ↪ ancestor
9       parse_constraint(line): add the constraint to the current node's constraints.
10  get_dependency_graph(function_graph) -> dependency_graph: taking the function graph, create a copy where all nodes without
    ↪  constraints also depend on their parents unless the target language implicitly tests all functions.
11  identify_strongly_connected_components(dependency_graph): return SCCs of the dependency graph and the edges between the
    ↪ SCCs.
12  compile_scc(scc, scc_graph, allow_autofill, allow_autodecomp): accumulate a implementation string which solves the current
    ↪ function
13    compile_children(scc, scc_graph, allow_autofill, allow_autodecomp): compile any SCCs this SCC depends on and add them to
      ↪ the implementation string.
14      compile_scc
15      direct_product_implementations(fn_implementations): return the direct product of the list of lists of
        ↪ fn_implementations
16    generate_implementations(scc, n, children_implementation_str): for each function in the SCC, generate n implementations
      ↪ of each function starting with the implementation string of the SCC's children.
17      fn_implementation
18    fn_implementation(fn_node, n): prompt the language model to generate n implementations of a function
19      generate_prompt(fn_node): first prepend a string with all descriptions, names, arguments, and returns of fn_node's
        ↪ direct children, in a style idiomatic for the target language. then, add fn_node's description and function
        ↪ signature.
20    solve_constraints(scc, fn_implementations, n, allow_autofill, allow_autodecomp): taking the provided constraints of each
      ↪ function in the scc, evaluate a shuffled list of the direct product of implementations with the constraints until
      ↪ one passes all of them
21      generate_constraints(fn_node): translate each of the constraints into an evaluation string idiomatic to the target
        ↪ language
22      eval_str(scc, implementation_str, allow_autofill): evaluate an implementation including constraints by running it in a
        ↪ target-language executor. if allow_autofill and the execution fails due to an undefined reference, attempt
        ↪ autofill
23        exec_implementation(implementation_str): run the implementation, including constraints/tests, in a target-language-
          ↪ specific executor, returning whether it was successful
24        attempt_autofill(scc, implementation_str, undefined_fn_use_example): create a new function node for the referenced
          ↪ function, then re-attempt to execute autofill
25          add_undefined_fn(scc, implementation_str, undefined_fn_caller, undefined_fn_use_example): create a new function
            ↪ node for the undefined function as a child of the function which calls it and add it to the scc and
            ↪ implementation string. prompt the language model with the usage example as the description to generate a set
            ↪ of implementations.
26            fn_implementation
27            eval_str
28    on_fail(scc, scc_graph, allow_autofill, allow_autodecomp): if allowing autodecomposition, attempt to decompose.
      ↪ otherwise, raise an error highlighting the scc which could not be compiled
29      attempt_autodecomp(scc, scc_graph, allow_autofill, allow_autodecomp): prompt the language model to decompose each
        ↪ unimplemented function node.
30        prompt_model(fn_node): prompt the language model, asking it to generate a "fn_name(arg): desc" for each subfunction
          ↪  necessary to implement the function node. add those functions to the scc, including a set of possible
          ↪ implementations for each.
31          fn_implementation
32          compile_scc
33      raise_error(scc): raise an error that Parsel could compile the scc
```

Figure A.12: Longer pseudocode of Parsel, including automatic infilling and automatic decomposition.

**Algorithm 1:** Parsel program synthesis – corresponds to short Parsel pseudocode.

---

**Input:** A string $program$ specifying a Parsel program and a $target\_language$
**Output:** Synthesized program
**Function** Parsel($program, target\_language$):
    $function\_graph \leftarrow$ ParseProgram($program$)
    $dependency\_graph \leftarrow$ GetDependencyGraph($function\_graph, target\_language$)
    $sccs, scc\_graph \leftarrow$ IdentifySCCs($dependency\_graph$)
    $implementation \leftarrow$ ""
    **foreach** $scc \in PostOrderTraversal(sccs)$ **do**
        $implementation \leftarrow implementation +$ SynthesizeSCC($scc, scc\_graph$)
    **return** $implementation$
**Function** ParseProgram($program$):
    $root \leftarrow CreateRootNode()$
    $current\_node \leftarrow root$
    $current\_indentation \leftarrow 0$
    **foreach** $line \in program$ **do**
        $line\_indentation \leftarrow$ IndentationOf($line$)
        **while** $line\_indentation < current\_indentation$ **do**
            $current\_node \leftarrow current\_node.parent$
            $current\_indentation -= 1$
        $current\_node \leftarrow$ ParseLine($line, current\_node$)
    **return** $root$
**Function** ParseLine($line, current\_node$):
    **if** *IsDefinition(line)* **then**
        current_node.children $\overset{+}{\leftarrow}$ ParseDefinition($line$);        *// Creates a new function node*
    **else**
        **if** *IsReference(line)* **then**
            current_node.children $\overset{+}{\leftarrow}$ ParseReference($line$);        *// Adds reference to current node*
        **else**
            current_node.constraints $\overset{+}{\leftarrow}$ ParseConstraint($line$);        *// Adds constraint to current node*
**Function** GetDependencyGraph($function\_graph, target\_language$):
    *// Converts function call graph into dependency graph.*
    $dependency\_graph \leftarrow$ CopyOf($function\_graph$);
    **foreach** $node \in dependency\_graph$ **do**
        **foreach** $child \in node.children$ **do**
            **if** $child.asserts = \emptyset$ **then**
                AddDependency($child, node$);        *// Child without asserts depends on parent*
    **return** $dependency\_graph$
**Function** SynthesizeSCC($scc, scc\_graph$):
    $implementation\_str \leftarrow$ SynthesizeChildren($scc, scc\_graph$)
    $fn\_implementations \leftarrow$ GenerateImplementations($scc, n, implementation\_str$)
    $implementation \leftarrow$ SolveConstraints($scc, fn\_implementations$)
    **if** *implementation is valid* **then**
        **return** $implementation$
**Function** SynthesizeChildren($scc, scc\_graph$):
    $child\_implementations \leftarrow$ ""
    **foreach** $child\_scc \in scc\_graph.GetChildren(scc)$ **do**
        $child\_implementations \leftarrow child\_implementations +$ SynthesizeSCC($child\_scc, scc\_graph$)
    **return** $child\_implementations$
**Function** GenerateImplementations($scc, n, implementation\_str$):
    $implementations \leftarrow []$
    **foreach** $fn \in scc$ **do**
        $fn\_implementations \leftarrow$ GenerateNImplementations($fn, n, implementation\_str$)
        $implementations$.append($fn\_implementations$)
    **return** $implementations$
**Function** SolveConstraints($scc, fn\_implementations$):
    $combinations \leftarrow$ DirectProductImplementations($fn\_implementations$)
    $constraints \leftarrow$ GenerateConstraints($scc$) ;        *// Translate each of the constraints into an evaluation string*
    **foreach** $implementation\_str \in combinations$ **do**
        **if** EvalStr($implementation\_str, constraints$) **then**
            **return** $implementation\_str$ ;        *// Return if implementation passes constraints*
    OnFail($scc, scc\_graph$) ;        *// If none are found, raise error*
**Function** GenerateConstraints($scc$):
    $constraints \leftarrow []$
    **foreach** $fn \in scc$ **do**
        $fn\_constraints \leftarrow$ GetConstraints($fn$)
        $constraints$.extend($fn\_constraints$)
    **return** $constraints$

---

## H Parsel Overview (Detailed)

We include a more detailed figure outlining Parsel.

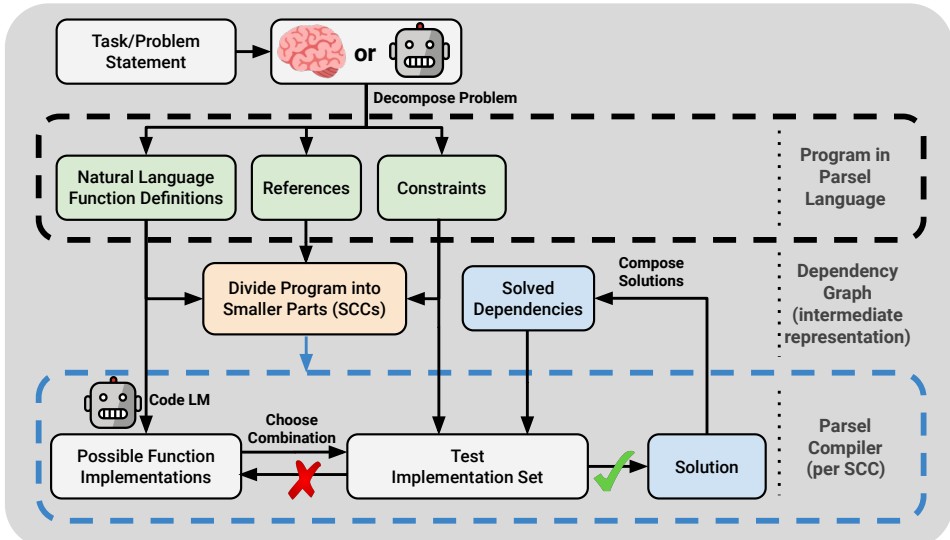

Figure A.13: **Parsel overview (detailed)**.

## I Lisp Interpreter

We include the Parsel code for a minimal Lisp interpreter.

```
1  An env is a dictionary of {'var':val} pairs, with a link to its outer environment in env['_outer'].
2  A procedure is a lambda expression, with parms, body, and env which calls eval_exp on the body.
3  #*#*#
4  evaluate_program(program): Initialize a standard environment. Parse and evaluate a list of expressions, returning the final
     ↪ result.
5  ['(define square (lambda (r) (* r r)))', '(square 3)'] -> 9
6    get_env(parms, args, env=None): Return a new env inside env with parms mapped to their corresponding args, and env as the
       ↪ new env's outer env.
7    [], [] -> {'_outer': None}
8    ['a'], [1] -> {'a': 1, '_outer': None}
9    standard_env(includes=['math','ops','simple_math']): An environment with some Scheme standard procedures. Start with an
       ↪ environment and update it with standard functions.
10   [] -> {'_outer': None}
11     get_math(): Get a dictionary mapping math library function names to their functions.
12     get_ops(): Get a dictionary mapping operator symbols to their functions: +, -, *, /, >, <, >=, <=, =.
13     get_simple_math(): Get a dictionary mapping 'abs', 'min', 'max', 'not', 'round' to their functions.
14     apply_fn_dict_key(fn_dict_generator, key, args_list): Return the value of fn_dict_generator()[key](*args_list) in
         ↪ standard_env.
15     get_math, 'sqrt', [4] -> 2.0
16     get_ops, '+', [1, 2] -> 3
17     get_simple_math, 'abs', [-1] -> 1
18       get_math
19       get_ops
20       get_simple_math
21   parse_and_update(expression, env): Parse an expression, return the result.
22   "(+ 1 (* 2 3))", {'+': (lambda x, y: x + y), '*': (lambda x, y: x * y), '_outer': None} -> 7
23     eval_exp(x, env): Evaluate an expression in an environment and return the result. Check if x is a list, a string, or
         ↪ neither, and call the corresponding function.
24     1, {'_outer': None} -> 1
25       find(env, var): Find the value of var in the innermost env where var appears.
26       {'a':4, '_outer':None}, 'a' -> 4
27       {'_outer':{'a':4, '_outer':None}}, 'a' -> 4
28       {'a':3, '_outer':{'a':4, '_outer':None}}, 'a' -> 3
29       string_case(x, env): Return find(env, x).
30       'a', {'a':4, '_outer':None} -> 4
31         find
32       list_case(x, env): Handle the function specified by the first value of x. Handle the first value of x being quote, if,
         ↪ define, set!, lambda, or otherwise. Return the result.
33       ['quote', 'a'], {'_outer': None} -> 'a'
34       ['if', True, 1, 2], {'_outer': None} -> 1
35       ['define', 'a', 1], {'_outer': None} -> None
36         get_procedure(parms, body, env): Return a procedure which evaluates body in a new environment with parms bound to the
           ↪ args passed to the procedure (in the same order as parms).
37           eval_procedure(parms, body, env, args): Gets a procedure and returns the result of evaluating proc(*args) in env.
             ↪ Should not be called directly.
38           ['r'], ['*', 'pi', ['*', 'r', 'r']], {'*': (lambda x, y: x * y), 'pi': 3, '_outer': None}, [1] -> 3
39             get_procedure
40             get_env
41             eval_exp
42         otherwise_case(x, env): Get the procedure by evaluating the first value of x. Then, evaluate the arguments and apply
           ↪ the procedure to them. Return the result.
43         ['+', 1, 2], {'+': (lambda x, y: x + y), '_outer': None} -> 3
44           eval_exp
45         eval_exp
46     not_list_case(x, env): Return x
47     1, {} -> 1
48   parse(program): Read a Scheme expression from a string.
49   '(1 + (2 * 3))' -> [1, '+', [2, '*', 3]]
50     tokenize(s): Convert a string into a list of tokens, including parens.
51     "1 + 2" -> ['1', '+', '2']
52     "1 + (2 * 3)" -> ['1', '+', '(', '2', '*', '3', ')']
53     read_from_tokens(tokens): Translate tokens to their corresponding atoms, using parentheses for nesting lists.
54     ['(', '1', '+', '(', '2', '*', '3', ')', ')'] -> [1, '+', [2, '*', 3]]
55       atom(token): Numbers become numbers; every other token is a string.
56       "1" -> 1
57       "a" -> "a"
58       "1.2" -> 1.2
59     nested_list_to_str(exp): Convert a nested list into a string with nesting represented by parentheses.
60     1 -> "1"
61     [1, '+', [2, '*', 3]] -> "(1 + (2 * 3))"
```

Figure A.14: Full Lisp interpreter implementation in Parsel, including constraints.

## J  Case Study

We include a simple example function we could not generate with Codex [12] directly from the top-level description in Figure A.15. The corresponding Python code (included in the appendix) is exactly 58 non-whitespace lines of code, including 17 lines of comments (3 corresponding to the descriptions), 2 asserts, and 39 lines implementing the three functions described as well as an automatically generated `get_number_of_active_cells_around_cell` function. In fact, using automatic decomposition, as discussed in Subsection R.3, it is not necessary to provide any of the function descriptions besides the top one. The model is (unsurprisingly) able to understand that `game_of_life_inversion_iteration` can be broken down into `invert_array` and `game_of_life_iteration`.

```
1 game_of_life_inversion_iteration(array_at_time_t): Takes a board and returns the next
    ↪ iteration of the game of life, but with all values flipped
2 [[0,0,1],[1,0,0],[1,0,0]] -> [[1,1,1],[1,0,1],[1,1,1]]
3 [[0,1,0,0],[1,0,1,0],[1,0,0,1],[0,1,1,0]] -> [[1,0,1,1],[0,1,0,1],[0,1,1,0],[1,0,0,1]]
4    game_of_life_iteration(array_at_time_t) -> array_at_time_t_plus_1: Takes a board with
       ↪ active and inactive cells as a list of lists and returns the next iteration of the
       ↪ game of life
5    array_inversion(array) -> inverted_array: Invert a square array by flipping 0's and 1's
```

Figure A.15: An example Parsel program for Python that takes in a list of lists representing a state of Conway's game of life [24] and returns the next state, with all the values inverted.

## K  Parsel Prompts

```
1 # Description: given a list of lists representing the cost of each edge, return an adjacency
    ↪ matrix corresponding to the minimum spanning true.
2 def minimum_spanning_tree(cost_matrix):
```

Figure A.16: Codex Prompt for an example leaf node

```
1 # Description: given a list of lists representing the cost of building a road between any two
    ↪  cities, and a list representing the cost of building an airport in a city, return a new
    ↪  cost matrix with a new node corresponding to the sky.
2 # Signature: sky_city_cost(city_road_cost, city_airport_cost)
3 from helpers import sky_city_cost
4
5 # Description: given a list of lists representing the cost of each edge, return an adjacency
    ↪ matrix corresponding to the minimum spanning true.
6 # Signature: minimum_spanning_tree(cost_matrix)
7 from helpers import minimum_spanning_tree
8
9 # Description: given a list of lists representing an adjacency matrix, return a list of the
    ↪ nodes connected to the final node. However, if only one node is connected to the final
    ↪ node, return an empty list.
10 # Signature: final_node_connectors(adjacency_matrix)
11 from helpers import final_node_connectors
12
13 # Description: given a matrix representing the cost of building a road between any two cities,
    ↪  and a list representing the cost of building an airport in a city (where any two cities
    ↪  with airports are connected), return a list of the cities that should have airports
    ↪ built in them to minimize the total cost of building roads and airports such that all
    ↪ cities are connected. The list should be sorted in ascending order.
14 # Uses: sky_city_cost, minimum_spanning_tree, final_node_connectors
15 def select_airport_cities(city_road_cost, city_airport_cost):
```

Figure A.17: Codex Prompt for an example merge node

```
1  # Reviewer:
2  # Please explain the above function in one sentence with as much detail as possible.
3  # In your one-sentence description, specify the range and domain of your function precisely.
4  # Your description should be clear enough that someone could reimplement the function from it.
     ↪
5  # Author:
6  # Sounds good, here's my one-sentence explanation of {name}:
7  # {name}
```

Figure A.18: Prompt format to generate descriptions for backtranslation

# L APPS Backtranslation

## L.1 Backtranslation / decompiling.

We anticipate that there are many programs that LLMs can implement by first generating Parsel code. But, as Parsel is a new framework, while language models can sometimes generate Parsel programs with few-shot prompts, it is not a syntax they have previously encountered. Thus, we may want to use existing code in other languages to construct datasets of Parsel programs from other languages. This requires us to first extract the call graph from the code, generate descriptions for each of the functions, and then generate Parsel programs from the graph. This call graph representation is convenient, so it is useful to have a bidirectional method to produce a graph from Parsel code and to produce Parsel code from the graph.

We filter the dataset to problems with starter code (providing the name of the evaluated function) and unit tests (provided as input-output pairs). For those tasks, we select solutions that define and call at least three functions, with at least one over 4 lines long and none over 15 lines.

As a proof of concept, we show 10 Parsel solutions which we could automatically generate from the APPS solutions. We generated the descriptions by prompting Codex to explain each function and its inputs and outputs. From this, we use backtranslation to attempt to implement these solutions in Python. We then verify that they are correct by applying the original unit tests as constraints on the root function. As mentioned in Section 1, the Parsel code is substantially shorter in terms of lines of code. We include these in Appendix L.

## L.2 Examples

We exclude the asserts in these examples for brevity - they correspond to those in the original dataset.

```
1 longest_palindrome(s): longest_palindrome takes a string s and returns the longest palindrome
  ↪  in s.
2  is_palindrome(s): is_palindrome returns True if the string s is the same forwards and
   ↪ backwards, and False otherwise.
3  check(li, ri, s): check takes a string s, a left index li, and a right index ri, and
   ↪ returns the longest palindrome that starts at or before li and ends at or after ri.
4   is_palindrome
```

Figure A.19: Train Problem 1638, Solution 2

```
1  # longest_palindrome takes a string s and returns the longest palindrome in s.
2  def longest_palindrome(s):
3      if len(s) <= 1:
4          return s
5      else:
6          longest = s[0]
7          for i in range(len(s)):
8              for j in range(len(s)):
9                  if is_palindrome(check(i, j, s)) and len(check(i, j, s)) > len(longest):
10                     longest = check(i, j, s)
11         return longest
12
13 # is_palindrome returns True if the string s is the same forwards and backwards, and False
   ↪ otherwise.
14 def is_palindrome(s):
15     if len(s) <= 1:
16         return True
17     else:
18         return s[0] == s[-1] and is_palindrome(s[1:-1])
19
20 # check takes a string s, a left index li, and a right index ri, and returns the longest
   ↪ palindrome that starts at or before li and ends at or after ri.
21 def check(li, ri, s):
22     while li >= 0 and ri < len(s) and s[li] == s[ri]:
23         li -= 1
24         ri += 1
25     return s[li+1:ri]
```

Figure A.20: Train Problem 1638, Solution 2

```
1  case_id(c_str): case_id takes a string and returns a string that is either "kebab", "snake",
   ↪ "camel", or "none" depending on whether the input string is in kebab case, snake case,
   ↪ camel case, or none of the above.
2  is_snake(s): is_snake takes a string and returns True if the string is snake_case and False
   ↪ otherwise.
3  is_kebab(s): is_kebab takes a string and returns True if the string is a kebab-case string,
   ↪ and False otherwise.
4  is_camel(s): is_camel returns True if the string s is not lowercase, does not contain
   ↪ dashes, and does not contain underscores.
```

Figure A.21: Train Problem 2892, Solution 7

```
1 # case_id takes a string and returns a string that is either "kebab", "snake", "camel", or "
  ↪ none" depending on whether the input string is in kebab case, snake case, camel case, or
  ↪  none of the above.
2 def case_id(c_str):
3     if is_snake(c_str) == True:
4         return "snake"
5     elif is_kebab(c_str) == True:
6         return "kebab"
7     elif is_camel(c_str) == True:
8         return "camel"
9     else:
10        return "none"
11
12 # is_snake takes a string and returns True if the string is snake_case and False otherwise.
13 def is_snake(s):
14     if s[0].isalpha() and s[0].islower() and len(s) > 1:
15         for char in s:
16             if char.isalpha():
17                 if char.isupper():
18                     return False
19             elif char == '_':
20                 pass
21             else:
22                 return False
23         return True
24     else:
25         return False
26
27 # is_kebab takes a string and returns True if the string is a kebab-case string, and False
  ↪ otherwise.
28 def is_kebab(s):
29     # if s is empty, False
30     if s == '':
31         return False
32     # if s is not a string, False
33     if type(s) != str:
34         return False
35     # if s is not lowercase, False
36     if s != s.lower():
37         return False
38     # if s contains anything other than a-z or -, False
39     for c in s:
40         if not (c.isalpha() or c == '-'):
41             return False
42     # if s contains a - at the beginning or end, False
43     if s[0] == '-' or s[-1] == '-':
44         return False
45     # if s contains more than one - in a row, False
46     for i in range(len(s)-1):
47         if s[i] == '-' and s[i+1] == '-':
48             return False
49     # otherwise, True
50     return True
51
52
53 # is_camel returns True if the string s is not lowercase, does not contain dashes, and does
  ↪ not contain underscores.
54 def is_camel(s):
55     return s != s.lower() and s.find('_') == -1 and s.find('-') == -1
```

Figure A.22: Train Problem 2892, Solution 7

```
1 find_2nd_largest(arr): find_2nd_largest takes a list of numbers and returns the second
  ↪ largest number in the list.
2  filter_int(arr): filter_int takes an array of integers and returns an array of integers.
3  sec_big(a, b): sec_big takes two numbers and returns the smaller of the two.
4  sort(arr): sort takes an array of numbers and returns a sorted array of numbers.
5  is_diff(arr): is_diff takes an array of numbers and returns True if there are any two
   ↪ numbers in the array that are different, and False if all the numbers in the array are
   ↪ the same.
```

Figure A.23: Train Problem 3090, Solution 9

```python
1  # find_2nd_largest takes a list of numbers and returns the second largest number in the list.
2  def find_2nd_largest(arr):
3      if len(arr) == 1:
4          return None
5      arr = filter_int(arr)
6      if len(arr) == 0:
7          return None
8      if len(arr) == 1:
9          return arr[0]
10     arr = sort(arr)
11     if is_diff(arr) == False:
12         return None
13     res = arr[len(arr) - 2]
14     return res
15
16 # filter_int takes an array of integers and returns an array of integers.
17 def filter_int(arr):
18     return list(filter(lambda x: type(x) == int, arr))
19
20 # sec_big takes two numbers and returns the smaller of the two.
21 def sec_big(a, b):
22     if a > b:
23         return b
24     else:
25         return a
26
27 # sort takes an array of numbers and returns a sorted array of numbers.
28 def sort(arr):
29     return sorted(arr)
30
31 # is_diff takes an array of numbers and returns True if there are any two numbers in the
   ↪ array that are different, and False if all the numbers in the array are the same.
32 def is_diff(arr):
33     for i in range(len(arr) - 1):
34         if arr[i] != arr[i + 1]:
35             return True
36     return False
```

Figure A.24: Train Problem 3090, Solution 9

```
1 happy_numbers(n): happy_numbers takes a positive integer n and returns a list of all the
  ↪ happy numbers between 1 and n, inclusive.
2  _is_happy_number(number): _is_happy_number takes a positive integer and returns True if the
   ↪  number is a happy number, False otherwise.
3    _sum_squares(number): _sum_squares takes a non-negative integer and returns the sum of
     ↪ the squares of its digits.
```

Figure A.25: Train Problem 3140, Solution 7

```
1  # happy_numbers takes a positive integer n and returns a list of all the happy numbers
   ↪ between 1 and n, inclusive.
2  def happy_numbers(n):
3      return [i for i in range(1, n + 1) if _is_happy_number(i)]
4
5  # _is_happy_number takes a positive integer and returns True if the number is a happy number,
   ↪ False otherwise.
6  def _is_happy_number(number):
7      # We want to make sure that the number is positive
8      if number < 0:
9          return False
10     # We want to make sure that the number is not 1
11     if number == 1:
12         return True
13     # We want to keep track of the numbers we have used
14     used_numbers = []
15     # We want to loop through the number
16     while number not in used_numbers:
17         # We want to add the number to the list of used numbers
18         used_numbers.append(number)
19         # We want to find the sum of the squares of the digits of the number
20         number = _sum_squares(number)
21         # We want to check if the number is one
22         if number == 1:
23             # We want to return True
24             return True
25     # We want to return False
26     return False
27
28 # _sum_squares takes a non-negative integer and returns the sum of the squares of its digits.
29 def _sum_squares(number):
30     if number < 0:
31         raise ValueError
32     else:
33         number = str(number)
34         sum = 0
35         for i in number:
36             sum += int(i) ** 2
37         return sum
```

Figure A.26: Train Problem 3140, Solution 7

```
1  am_i_wilson(n): am_i_wilson(n) returns True if n is a prime number between 2 and 563,
   ↪ inclusive, and False otherwise.
2  is_prime(n): is_prime takes a positive integer n and returns True if n is prime and False
   ↪ otherwise.
3  factorial(n): factorial(n) returns the product of all integers from 1 to n, inclusive.
```

Figure A.27: Train Problem 3229, Solution 26

```
1 # am_i_wilson(n) returns True if n is a prime number between 2 and 563, inclusive, and False
    ↪ otherwise.
2 def am_i_wilson(n):
3     if is_prime(n) and 2 <= n and n <= 563:
4         return (factorial(n-1) + 1) % (n**2) == 0
5     else:
6         return False
7
8 # is_prime takes a positive integer n and returns True if n is prime and False otherwise.
9 def is_prime(n):
10     if n == 2:
11         return True
12     if n == 3:
13         return True
14     if n % 2 == 0:
15         return False
16     if n % 3 == 0:
17         return False
18     i = 5
19     w = 2
20     while i * i <= n:
21         if n % i == 0:
22             return False
23         i += w
24         w = 6 - w
25     return True
26
27
28 # factorial(n) returns the product of all integers from 1 to n, inclusive.
29 def factorial(n):
30     if n == 0:
31         return 1
32     else:
33         return n * factorial(n-1)
```

Figure A.28: Train Problem 3229, Solution 26

```
1 am_i_wilson(n): am_i_wilson takes a positive integer n and returns True if n is prime and (n
    ↪ -1)! + 1 is divisible by n^2, and False otherwise.
2  fac(n): fac is a function that takes a positive integer n and returns the product of all
    ↪ integers from 1 to n.
3  is_prime(n): is_prime takes a positive integer n and returns True if n is prime and False
    ↪ otherwise.
```

Figure A.29: Train Problem 3229, Solution 71

```
1  # am_i_wilson takes a positive integer n and returns True if n is prime and (n-1)! + 1 is
   ↪ divisible by n^2, and False otherwise.
2  def am_i_wilson(n):
3      return is_prime(n) and (fac(n-1) + 1) % n**2 == 0
4
5  # fac is a function that takes a positive integer n and returns the product of all integers
   ↪ from 1 to n.
6  def fac(n):
7      if n == 0:
8          return 1
9      return n * fac(n-1)
10 # is_prime takes a positive integer n and returns True if n is prime and False otherwise.
11 def is_prime(n):
12     if n == 2:
13         return True
14     elif n < 2 or n % 2 == 0:
15         return False
16     for i in range(3, int(n**0.5)+1, 2):
17         if n % i == 0:
18             return False
19     return True
```

Figure A.30: Train Problem 3229, Solution 71

```
1  evil(n): evil(n) returns "It's Evil!" if n is an evil number, otherwise it returns "It's
   ↪ Odious!" The range of evil is the set of all integers, and the domain is the set of all
   ↪ strings.
2  evilometer(n): evilometer(n) is a generator that yields n times if n is even, and yields n
   ↪ // 2 times if n is odd.
```

Figure A.31: Train Problem 3321, Solution 33

```
1  # evil(n) returns "It's Evil!" if n is an evil number, otherwise it returns "It's Odious!"
   ↪ The range of evil is the set of all integers, and the domain is the set of all strings.
2  def evil(n):
3      gen = evilometer(n)
4      if sum(list(gen)) % 2 == 0:
5          return "It's␣Evil!"
6      else:
7          return "It's␣Odious!"
8
9  # evilometer(n) is a generator that yields n times if n is even, and yields n // 2 times if n
   ↪  is odd.
10 def evilometer(n):
11     while n:
12         yield n
13         if n % 2:
14             n //= 2
15         else:
16             n -= 1
```

Figure A.32: Train Problem 3321, Solution 33

```
1  circular_prime(number): circular_prime takes a number and returns True if it is a circular
   ↪ prime, and False otherwise.
2  rotate(l, n): rotate takes a list and an integer and returns a new list with the last n
   ↪ elements of the original list moved to the front.
3  is_prime(n): is_prime returns True if n is a prime number, and False otherwise.
```

Figure A.33: Train Problem 3368, Solution 5

```
1  # circular_prime takes a number and returns True if it is a circular prime, and False
   ↪ otherwise.
2  def circular_prime(number):
3      if is_prime(number) == False:
4          return False
5      else:
6          for i in range(len(str(number))-1):
7              number = rotate(str(number), 1)
8              if is_prime(int(number)) == False:
9                  return False
10         return True
11
12 # rotate takes a list and an integer and returns a new list with the last n elements of the
   ↪ original list moved to the front.
13 def rotate(l, n):
14     return l[-n:] + l[:-n]
15
16 # is_prime returns True if n is a prime number, and False otherwise.
17 def is_prime(n):
18     if n < 2:
19         return False
20     for i in range(2, int(n ** 0.5) + 1):
21         if n % i == 0:
22             return False
23     return True
```

Figure A.34: Train Problem 3368, Solution 5

```
1  circular_prime(n): circular_prime(n) returns True if n is a circular prime, and False
   ↪ otherwise.
2   is_prime(n): is_prime takes a number n and returns True if n is prime and False if n is not
    ↪ prime.
3   circul_num(n, l): circul_num takes an integer n and an integer l, and returns the integer
    ↪ that results from moving the first digit of n to the end of n, and then padding the
    ↪ result with zeros until it has length l.
```

Figure A.35: Train Problem 3368, Solution 6

```
1  # circular_prime(n) returns True if n is a circular prime, and False otherwise.
2  def circular_prime(n):
3      # Check if n is a prime number.
4      if not is_prime(n):
5          return False
6      else:
7          # Get the length of n.
8          length = len(str(n))
9          # Get the first digit of n.
10         first_digit = str(n)[0]
11         # Get the number of rotations.
12         rotations = length
13         # Rotate n.
14         while rotations > 0:
15             # Get the new number.
16             n = circul_num(n, length)
17             # Check if n is a prime number.
18             if not is_prime(n):
19                 return False
20             # Decrement the number of rotations.
21             rotations -= 1
22         return True
23 # is_prime takes a number n and returns True if n is prime and False if n is not prime.
24 def is_prime(n):
25     if n == 2 or n == 3:
26         return True
27     if n % 2 == 0 or n < 2:
28         return False
29     for i in range(3, int(n**0.5)+1, 2): # only odd numbers
30         if n % i == 0:
31             return False
32     return True
33
34 # circul_num takes an integer n and an integer l, and returns the integer that results from
   ↪ moving the first digit of n to the end of n, and then padding the result with zeros
   ↪ until it has length l.
35 def circul_num(n, l):
36     n = str(n)
37     digit = n[0]
38     n = n[1:] + digit
39     n = n.ljust(l, "0")
40     return int(n)
```

Figure A.36: Train Problem 3368, Solution 6

```
1  list_squared(m, n): list_squared(m, n) returns a list of lists of integers, where each
   ↪ sublist contains an integer and its sum of squared divisors, and the integers are in the
   ↪  range [m, n] (inclusive).
2   divisors_list(num): divisors_list takes a number and returns a list of all the divisors of
    ↪ that number.
3   sum_squares(nums): sum_squares takes a list of numbers and returns the sum of the squares
    ↪ of those numbers.
4   isSquarable(num): isSquarable takes a number and returns True if the number is a perfect
    ↪ square, and False otherwise.
```

Figure A.37: Train Problem 3393, Solution 35

```python
1  # list_squared(m, n) returns a list of lists of integers, where each sublist contains an
       ↪ integer and its sum of squared divisors, and the integers are in the range [m, n] (
       ↪ inclusive).
2  def list_squared(m, n):
3      result = []
4      for num in range(m, n + 1):
5          divisors = divisors_list(num)
6          sum_of_squares = sum_squares(divisors)
7          if isSquarable(sum_of_squares):
8              result.append([num, sum_of_squares])
9      return result
10 # divisors_list takes a number and returns a list of all the divisors of that number.
11 def divisors_list(num):
12     divisors = []
13     for i in range(1, num + 1):
14         if num % i == 0:
15             divisors.append(i)
16     return divisors
17
18
19 # sum_squares takes a list of numbers and returns the sum of the squares of those numbers.
20 def sum_squares(nums):
21     sum = 0
22     for i in range(len(nums)):
23         sum += nums[i]**2
24     return sum
25
26 # isSquarable takes a number and returns True if the number is a perfect square, and False
       ↪ otherwise.
27 def isSquarable(num):
28     root = num ** (1/2)
29     if root % 1 == 0:
30         return True
31     else:
32         return False
```

Figure A.38: Train Problem 3393, Solution 35

# M    Compiled Examples

```python
1   # Takes a board and returns the next iteration of the game of life, but with all values
    ↪ flipped
2   def game_of_life_inversion_iteration(array_at_time_t):
3       # Your code here
4       #return game_of_life_iteration(invert_array(array_at_time_t))
5       return invert_array(game_of_life_iteration(array_at_time_t))
6
7   # Takes a board and returns the next iteration of the game of life
8   def invert_array(array_at_time_t):
9       return [list(map(lambda x: 1-x, row)) for row in array_at_time_t]
10
11  # Takes a board and returns the board with all values flipped
12  def game_of_life_iteration(array_at_time_t):
13      # The array that will be returned
14      array_at_time_t_plus_1 = []
15
16      # Iterate through the rows of the array
17      for i in range(0, len(array_at_time_t)):
18          # The array that will contain the next row
19          next_row = []
20
21          # Iterate through the columns of the array
22          for j in range(0, len(array_at_time_t[i])):
23              # The number of neighbors
24              num_neighbors = 0
25
26              # Iterate through the neighbors of the cell
27              for k in range(-1, 2):
28                  for l in range(-1, 2):
29                      # Don't count the cell itself
30                      if k == 0 and l == 0:
31                          continue
32
33                      # Check if the neighbor is valid
34                      if i + k >= 0 and i + k < len(array_at_time_t) and j + l >= 0 and j + l <
                        ↪ len(array_at_time_t[i]):
35                          # If the neighbor is alive, increment the number of neighbors
36                          if array_at_time_t[i + k][j + l] == 1:
37                              num_neighbors += 1
38
39              # If the cell is alive, check if it should die
40              if array_at_time_t[i][j] == 1:
41                  if num_neighbors < 2 or num_neighbors > 3:
42                      next_row.append(0)
43                  else:
44                      next_row.append(1)
45              # If the cell is dead, check if it should become alive
46              else:
47                  if num_neighbors == 3:
48                      next_row.append(1)
49                  else:
50                      next_row.append(0)
51
52          # Add the next row to the array
53          array_at_time_t_plus_1.append(next_row)
54
55      # Return the next array
56      return array_at_time_t_plus_1
57
58  assert game_of_life_inversion_iteration([[0, 0, 1], [1, 0, 0], [1, 0, 0]]) == [[1, 1, 1], [1,
    ↪ 0, 1], [1, 1, 1]]
59  assert game_of_life_inversion_iteration([[0, 1, 0, 0], [1, 0, 1, 0], [1, 0, 0, 1], [0, 1, 1,
    ↪ 0]]) == [[1, 0, 1, 1], [0, 1, 0, 1], [0, 1, 1, 0], [1, 0, 0, 1]]
```

Figure A.39: Compilation of the program taking Conway's game of life [24] and returning the next state, with the values inverted. In this case, we use automatic decomposition to generate the two subfunctions of `game_of_life_iteration`.

```
1  # Takes a board with active and inactive cells and returns the next iteration of the game of
   ↪ life, but with all the values flipped
2  def game_of_life_inversion_iteration(array_at_time_t):
3      return array_inversion(game_of_life_iteration(array_at_time_t))
4
5  # Takes a board with active and inactive cells as a list of lists and returns the next
   ↪ iteration of the game of life
6  def game_of_life_iteration(array_at_time_t):
7      array_at_time_t_plus_1 = []
8      for row in range(len(array_at_time_t)):
9          array_at_time_t_plus_1.append([])
10         for column in range(len(array_at_time_t[row])):
11             array_at_time_t_plus_1[row].append(0)
12     for row in range(len(array_at_time_t)):
13         for column in range(len(array_at_time_t[row])):
14             if array_at_time_t[row][column] == 1:
15                 if count_neighbors(array_at_time_t, row, column) < 2:
16                     array_at_time_t_plus_1[row][column] = 0
17                 elif count_neighbors(array_at_time_t, row, column) > 3:
18                     array_at_time_t_plus_1[row][column] = 0
19                 else:
20                     array_at_time_t_plus_1[row][column] = 1
21             else:
22                 if count_neighbors(array_at_time_t, row, column) == 3:
23                     array_at_time_t_plus_1[row][column] = 1
24     return array_at_time_t_plus_1
25
26 # Invert a square array by replacing all 0's with 1's and vice versa
27 def array_inversion(array):
28     inverted_array = []
29     for i in range(len(array)):
30         inverted_array.append([])
31         for j in range(len(array[i])):
32             inverted_array[i].append(1 - array[i][j])
33     return inverted_array
34
35 #
36 def count_neighbors(array_at_time_t, row, column):
37     count = 0
38     for i in range(row-1, row+2):
39         for j in range(column-1, column+2):
40             if i == row and j == column:
41                 continue
42             if i < 0 or j < 0:
43                 continue
44             if i >= len(array_at_time_t) or j >= len(array_at_time_t[0]):
45                 continue
46             if array_at_time_t[i][j] == 1:
47                 count += 1
48     return count
49
50
51 assert game_of_life_inversion_iteration([[0, 0, 1], [1, 0, 0], [1, 0, 0]]) == [[1, 1, 1], [1,
   ↪  0, 1], [1, 1, 1]]
52 assert game_of_life_inversion_iteration([[0, 1, 0, 0], [1, 0, 1, 0], [1, 0, 0, 1], [0, 1, 1,
   ↪  0]]) == [[1, 0, 1, 1], [0, 1, 0, 1], [0, 1, 1, 0], [1, 0, 0, 1]]
53
54 assert game_of_life_iteration([[0, 0, 1], [1, 0, 0], [1, 0, 0]]) == [[0, 0, 0], [0, 1, 0], [0,
   ↪  0, 0]]
55 assert game_of_life_iteration([[0, 1, 0, 0], [1, 0, 1, 0], [1, 0, 0, 1], [0, 1, 1, 0]]) ==
   ↪  [[0, 1, 0, 0], [1, 0, 1, 0], [1, 0, 0, 1], [0, 1, 1, 0]]
56
57 assert array_inversion([[1]]) == [[0]]
58 assert array_inversion([[0, 1], [1, 0]]) == [[1, 0], [0, 1]]
```

Figure A.40: Compilation of the program taking Conway's game of life [24] and returning the next state, with the values inverted. In this case, we use automatic infilling to generate the `count_neighbors` function.

```
1  # Calls base_case if 1, otherwise recursion_rule
2  def collatz_recursion(num, cur_list=list()):
3      if num == 1:
4          return base_case(num, cur_list)
5      else:
6          return recursion_rule(num, cur_list)
7
8  # Returns the list with the number appended to it
9  def base_case(num, cur_list):
10     cur_list.append(num)
11     return cur_list
12
13
14 # Add num to list, collatz with 3n + 1 if odd or n / 2 if even
15 def recursion_rule(num, cur_list):
16     cur_list.append(num)
17     if num % 2 == 0:
18         return collatz_recursion(num / 2, cur_list)
19     else:
20         return collatz_recursion((3 * num) + 1, cur_list)
21
22
23 assert collatz_recursion(19) == [19, 58, 29, 88, 44, 22, 11, 34, 17, 52, 26, 13, 40, 20, 10,
   ↪ 5, 16, 8, 4, 2, 1]
```

Figure A.41: Compilation of the program generating a list corresponding to the Collatz conjecture.

```
1  -- if p ∧ q, then q ∧ p
2  lemma p_q_implies_q_p(p q: Prop):
3
4      p ∧ q → q ∧ p :=
5  begin
6      intro h,
7      cases h with hp hq,
8      split,
9          exact hq,
10         exact hp,
11 end
12
13 -- Description: if p ∨ q, then q ∨ p
14 -- if q ∧ p, then p ∧ q
15 lemma q_p_implies_p_q(p q: Prop):
16
17     (q ∧ p) → (p ∧ q) :=
18 begin
19   intro h,
20   split,
21     exact h.right,
22     exact h.left,
23 end
24
25 /-
26   Theorem:
27     If q ∧ p, then p ∧ q
28 -/
29 -- the and operator is commutative
30 lemma and_commute(p q: Prop):
31   (p ∧ q → q ∧ p) ∧ (q ∧ p → p ∧ q) :=
32
33 begin
34   apply and.intro,
35   { apply p_q_implies_q_p },
36   { apply q_p_implies_p_q }
37 end
38
39 -- Description: if p ∧ q, then p
40 -- Signature: p_and_q_implies_p(p q: Prop)
41
42 -- show (p ∧ q → q ∧ p) ∧ (q ∧ p → p ∧ q)
```

Figure A.42: Generated proof of `and_commute`. We trim the post-proof comments elsewhere.

```
1  # given a list of lists representing the cost of building a road between any two cities, and a list representing the cost of
↪   building an airport in a city, return a new cost matrix with a new node corresponding to the sky.
2  def sky_city_cost(city_road_cost, city_airport_cost):
3      """
4      :param city_road_cost: list of lists representing cost of road between any two cities
5      :param city_airport_cost: list representing cost of an airport in a city
6      :return: new cost matrix with a new node corresponding to the sky
7      """
8      # add new node for sky to cost matrix
9      num_cities = len(city_road_cost)
10     sky_city_cost = [[0 for _ in range(num_cities + 1)] for _ in range(num_cities + 1)]
11     for i in range(num_cities):
12         for j in range(num_cities):
13             sky_city_cost[i][j] = city_road_cost[i][j]
14     for i in range(num_cities):
15         sky_city_cost[i][-1] = city_airport_cost[i]
16         sky_city_cost[-1][i] = city_airport_cost[i]
17     return sky_city_cost
18
19 # given a list of lists representing the cost of each edge, return an adjacency matrix corresponding to the minimum spanning
↪   true.
20 def minimum_spanning_tree(cost_matrix):
21     # This is a list of the vertices that have been added to the MST
22     visited = [0]
23     # This is a list of the vertices that have not been added to the MST
24     unvisited = [i for i in range(1, len(cost_matrix))]
25     # This is a list of edges that are part of the MST
26     edges = []
27     # This is the adjacency matrix corresponding to the MST
28     adjacency_matrix = [[0 for i in range(len(cost_matrix))] for j in range(len(cost_matrix))]
29     while len(unvisited) > 0:
30         # Get the index of the minimum edge
31         min_edge_index = -1
32         min_edge_value = float('inf')
33         for i in range(len(visited)):
34             for j in range(len(unvisited)):
35                 if cost_matrix[visited[i]][unvisited[j]] < min_edge_value:
36                     min_edge_index = (visited[i], unvisited[j])
37                     min_edge_value = cost_matrix[visited[i]][unvisited[j]]
38         # Add the minimum edge to our MST
39         edges.append(min_edge_index)
40         # Add the unvisited vertex to the list of visited vertices
41         visited.append(min_edge_index[1])
42         # Remove the unvisited vertex from the list of unvisited vertices
43         unvisited.remove(min_edge_index[1])
44     # Add edges to the adjacency matrix
45     for edge in edges:
46         adjacency_matrix[edge[0]][edge[1]] = 1
47         adjacency_matrix[edge[1]][edge[0]] = 1
48     return adjacency_matrix
49
50 # given a list of lists representing an adjacency matrix, return a list of the nodes connected to the final node. However,
↪   if only one node is connected to the final node, return an empty list.
51 def final_node_connectors(adjacency_matrix):
52     final_node = len(adjacency_matrix) - 1
53     final_node_connectors = []
54     for i in range(len(adjacency_matrix) - 1):
55         if adjacency_matrix[i][final_node] == 1:
56             final_node_connectors.append(i)
57     if len(final_node_connectors) == 1:
58         return []
59     return final_node_connectors
60
61 # given a matrix representing the cost of building a road between any two cities, and a list representing the cost of
↪   building an airport in a city (where any two cities with airports are connected), return a list of the cities that
↪   should have airports built in them to minimize the total cost of building roads and airports such that all cities are
↪   connected. The list should be sorted in ascending order.
62 def select_airport_cities(city_road_cost, city_airport_cost):
63     cost_matrix = sky_city_cost(city_road_cost, city_airport_cost)
64     adjacency_matrix = minimum_spanning_tree(cost_matrix)
65     return final_node_connectors(adjacency_matrix)
66
67 assert repr(str(select_airport_cities([[0, 3, 3], [3, 0, 3], [3, 3, 0]], [0, 0, 0]))) == repr(str([0, 1, 2]))
68 assert repr(str(select_airport_cities([[0, 3, 3], [3, 0, 3], [3, 3, 0]], [10, 10, 10]))) == repr(str([]))
69 assert repr(str(select_airport_cities([[0, 10, 3], [10, 0, 11], [3, 11, 0]], [1, 4, 5]))) == repr(str([0, 1]))
70
71 assert repr(str(sky_city_cost([[1, 2, 3], [1, 2, 3], [1, 2, 3]], [4, 5, 6]))) == repr(str([[1, 2, 3, 4], [1, 2, 3, 5], [1, 2,
↪   3, 6], [4, 5, 6, 0]]))
72
73 assert repr(str(minimum_spanning_tree([[0, 1, 3, 4], [1, 0, 2, 100], [3, 2, 0, 5], [4, 100, 5, 0]]))) == repr(str([[0, 1, 0,
↪   1], [1, 0, 1, 0], [0, 1, 0, 0], [1, 0, 0, 0]]))
74
75 assert repr(str(final_node_connectors([[0, 1, 0, 1], [1, 0, 1, 0], [0, 1, 0, 0], [1, 0, 0, 0]]))) == repr(str([]))
76 assert repr(str(final_node_connectors([[0, 1, 0, 1], [1, 0, 1, 0], [0, 1, 0, 1], [1, 0, 1, 0]]))) == repr(str([0, 2]))
```

Figure A.43: Generated solution for `select_airport_cities`.

```
1  # chess_game takes a board and returns the winner of the game.
2  def chess_game(board):
3      if weight(board) == 0:
4          return 'Draw'
5      if weight(board) > 0:
6          return 'White'
7      if weight(board) < 0:
8          return 'Black'
9
10 # weight takes a board and returns the weight of the board.
11 def weight(board):
12     weight = 0
13     for row in range(len(board)):
14         for col in range(len(board[row])):
15             if board[row][col] != '.':
16                 weight += weight_helper(board, row, col, 0)
17     return weight
18
19 # weight_helper takes a board, a row, a column, and a weight and returns the weight of the
   ↪ board.
20 def weight_helper(board, row, col, weight):
21     if board[row][col] == '.':
22         return 0
23     elif board[row][col] == 'Q':
24         return weight + 9
25     elif board[row][col] == 'R':
26         return weight + 5
27     elif board[row][col] == 'B':
28         return weight + 3
29     elif board[row][col] == 'N':
30         return weight + 3
31     elif board[row][col] == 'P':
32         return weight + 1
33     elif board[row][col] == 'q':
34         return weight - 9
35     elif board[row][col] == 'r':
36         return weight - 5
37     elif board[row][col] == 'b':
38         return weight - 3
39     elif board[row][col] == 'n':
40         return weight - 3
41     elif board[row][col] == 'p':
42         return weight - 1
43     else:
44         return weight
45
46 assert repr(str(chess_game('...QK...\n........\n........\n........\n........\n........\n
   ↪ ........\n...rk...'))) == repr('White')
47 assert repr(str(chess_game('rnbqkbnr\npppppppp\n........\n........\n........\n........\
   ↪ nPPPPPPPP\nRNBQKBNR'))) == repr('Draw')
48 assert repr(str(chess_game('rppppppr\n...k....\n........\n........\n........\n........\nK...Q
   ↪ ...\n........'))) == repr('Black')
49 assert repr(str(chess_game('....bQ.K\n.B......\n.....P..\n........\n........\n........\n...N.
   ↪ P..\n.....R..'))) == repr('White')
50 ...
```

Figure A.44: Generated solution for Problem 368 of the APPS test set, identifying the leader of a chess game from the board.

# N APPS Decomposition Prompts and Evaluation Hyperparameters

We slightly loosen the requirements for Parsel programs generated by language models, treating redundant function definitions as references instead of raising errors. We sample everything with temperature=0.6, except the translations which we sample with temperature=0.2, a presence penalty of 0.1, and a logit bias to prevent it from generating the text "def", as Codex has a tendency to degenerate to producing Python even when prompted with Parsel examples. We allow at most 500 tokens per function, but in practice found that they typically used less than half of them.

For evaluation, we use a timeout of 0.04 seconds per solution and evaluate at most 100,000 implementations per generated Parsel program.

For the Codex-only ablation, we allow it to generate at most 1000 tokens, in large part due to the rate limit. In particular, there is a heuristic rate limit that rejects any calls requesting more than 4,000 tokens. As a result, any larger number of samples per problem would prevent batching more than 3 samples at a time.

```
1 """An action plan is a list of strings that describes a sequence of steps to accomplish a
   ↪ task, To be correctly parsed, an action plan must be syntactically correct and contain
   ↪ only allowed actions and recognizable simple objects. Allowed actions: 'close' <arg1>, '
   ↪ cut' <arg1>, 'drink' <arg1>, 'drop' <arg1>, 'eat' <arg1>, 'find' <arg1>, 'grab' <arg1>, '
   ↪ greet' <arg1>, 'lie on' <arg1>, 'look at' <arg1>, 'open' <arg1>, 'plug in' <arg1>, 'plug
   ↪  out' <arg1>, 'point at' <arg1>, 'pour' <arg1> 'into' <arg2>, 'pull' <arg1>, 'push' <arg1
   ↪ >, 'put' <arg1> 'on' <arg2>, 'put' <arg1> 'in' <arg2>, 'put back' <arg1>, 'take off' <
   ↪ arg1>, 'put on' <arg1>, 'read' <arg1>, 'release', 'rinse' <arg1>, 'run to' <arg1>, '
   ↪ scrub' <arg1>, 'sit on' <arg1>, 'sleep', 'squeeze' <arg1>, 'stand up', 'switch off' <arg1
   ↪ >, 'switch on' <arg1>, 'touch' <arg1>, 'turn to' <arg1>, 'type on' <arg1>, 'wake up', '
   ↪ walk to' <arg1>, 'wash' <arg1>, 'watch' <arg1>, 'wipe' <arg1>. To satisfy the common-
   ↪ sense constraints, each action step in this action plan must not violate the set of its
   ↪ pre-conditions (e.g. the agent cannot grab milk from the fridge before opening it) and
   ↪ post-conditions (e.g. the state of the fridge changes from "closed" to "open" after the
   ↪ agent opens it)."""
2 #*#*#
3 task_plan(): return a list of strings that represents an action plan to put a mug on the
   ↪ stall and bread on the desk.
4 -> "executable"
5     put_object_on(object, place): return a list of strings that represents an action plan to
       ↪ put an object in a place.
6     "mug", "stall" -> "executable"
```

Figure A.45: Full Parsel program including header for Fig. 2 example, with the #*#*# as the header seperator. Note that we essentially just took the executability definition in [28] and added the list of available actions.

```
1 """-----Solution-----
2
3 Propose a clever and efficient high-level solution for this problem. Consider all edge cases
   ↪ and failure modes.
4
5 Some common strategies include:
6 Constructive algorithms, Binary search, Depth-first search (DFS) and similar algorithms,
   ↪ Dynamic programming, Bitmasks, Brute force, Greedy algorithms, Graphs, Two pointers,
   ↪ Trees, Geometry, Graph matchings, Hashing, Probabilities, Data structures, Sortings,
   ↪ Games, Number theory, Combinatorics, Divide and conquer, Disjoint set union (DSU),
   ↪ Expression parsing
7
8 Write out your reasoning first, and then describe your high-level solution and explain why it
   ↪  is correct.
9 \"\"\"
10 Let's think step by step to come up with a clever algorithm."""
```

Figure A.46: High-level sketch prompt for APPS programs

```
 1  """-----Translation-----
 2  # Here is an example calculating the probability of landing on the same character in a random shift of an input string, based on the following
    ↪  problem:
 3  # Vasya and Kolya play a game with a string, using the following rules. Initially, Kolya creates a string s, consisting of small English letters, and
    ↪   uniformly at random chooses an integer k from a segment [0, len(s) - 1]. He tells Vasya this string s, and then shifts it k letters to the left
    ↪   , i.e. creates a new string t = s_k + 1s_k + 2... s_ns_1s_2... s_k. Vasya does not know the integer k nor the string t, but he wants to guess
    ↪   the integer k. To do this, he asks Kolya to tell him the first letter of the new string, and then, after he sees it, open one more letter on
    ↪   some position, which Vasya can choose.
 4  # Vasya understands, that he can't guarantee that he will win, but he wants to know the probability of winning, if he plays optimally. He wants you
    ↪   to compute this probability.
 5  # Note that Vasya wants to know the value of k uniquely, it means, that if there are at least two cyclic shifts of s that fit the information Vasya
    ↪   knowns, Vasya loses. Of course, at any moment of the game Vasya wants to maximize the probability of his win.
 6  \"\"\"
 7  generate_cyclic_shifts(input_str): Calculates the average number of unique characters in the substrings of the input string that start with each
    ↪   character.
 8  parse_input(input_str): Takes a string and returns the input string
 9  compute_a_and_letter_pos(input_str): Generates the str_as_number_list and letter_pos lists. str_as_number_list is a list of integers that is used
    ↪   to store the character values of the input string. str_as_number_list is initialized as a list of 0s for twice the length of the input string.
    ↪   The values are calculated by taking the ASCII value of each character in the string and subtracting the ASCII value of the character 'a'.
    ↪   letter_pos is a list of lists, with each sublist containing the indices at which a particular character appears in the input string.
10  compute_unique_characters(c, str_as_number_list, letter_pos) -> ans: Calculates the maximum number of unique characters in all substrings (for k=1
    ↪   to length) that start with the character represented by c. letter_pos is a list of lists, with each sublist containing the indices at which a
    ↪   character appears in the input string. str_as_number_list is a list of integers that is used to store the character values of the input string
    ↪   .
11  compute_unique_characters_for_k(c, k, str_as_number_list, letter_pos): Create a counts list of zeros for each of the 26 alphabetical characters.
    ↪   For each i in the sublist of positions of letter_pos[c], increment counts at str_as_number_list[i + k]. Return the number of counts which
    ↪   are exactly one.
12  to_output_str(ans, input_str): Returns a string representation of ans divided by the length of the input string.
13  \"\"\"
14  (6 lines)
15
16  # And here is an example identifying the largest binary number according to the following rules:
17  # The Little Elephant has an integer a, written in the binary notation. He wants to write this number on a piece of paper.
18  # To make sure that the number a fits on the piece of paper, the Little Elephant ought to delete exactly one any digit from number a in the binary
    ↪   record. At that a new number appears. It consists of the remaining binary digits, written in the corresponding order (possible, with leading
    ↪   zeroes).
19  # The Little Elephant wants the number he is going to write on the paper to be as large as possible. Help him find the maximum number that he can
    ↪   obtain after deleting exactly one binary digit and print it in the binary notation.
20  \"\"\"
21  largest_binary_number(input_str): Returns the largest binary number that can be made by removing at most one digit from the input string.
22  parse_input(input_str): Takes a string and returns the input string
23  remove_zero(binary_str): Remove the first zero from the input string.
24  to_output_str(bigger_str): Returns the bigger string.
25  \"\"\"
26  (4 lines)
27
28  # Here is an example of the format applied to identifying the winner of the following game:
29  # It is so boring in the summer holiday, isn't it? So Alice and Bob have invented a new game to play. The rules are as follows. First, they get a set
    ↪    of n distinct integers. And then they take turns to make the following moves. During each move, either Alice or Bob (the player whose turn is
    ↪    the current) can choose two distinct integers x and y from the set, such that the set doesn't contain their absolute difference |x - y|. Then
    ↪    this player adds integer |x - y| to the set (so, the size of the set increases by one).
30  # If the current player has no valid move, he (or she) loses the game. The question is who will finally win the game if both players play optimally.
    ↪   Remember that Alice always moves first.
31  \"\"\"
32  identify_winner(input_str): Returns the winner of the game.
33  parse_input(input_str): Takes a string containing the length on the first line and the integers on the second and returns the list of integers
34  num_moves(l): The number of moves is the largest element in the list divided by the greatest common divisor of all elements in the list, minus the
    ↪   length of the list.
35    all_gcd(l): Returns the greatest common divisor of all elements in the list
36  to_output_str(num_moves): Returns the string 'Alice' if the number of moves is odd and 'Bob' if the number of moves is even
37  \"\"\"
38  (5 lines)
39
40  # Limak is a little bear who loves to play. Today he is playing by destroying block towers. He built n towers in a row. The i-th tower is made of h_i
    ↪    identical blocks. For clarification see picture for the first sample.
41  # Limak will repeat the following operation till everything is destroyed.
42  # Block is called internal if it has all four neighbors, i.e. it has each side (top, left, down and right) adjacent to other block or to the floor.
    ↪   Otherwise, block is boundary. In one operation Limak destroys all boundary blocks. His paws are very fast and he destroys all those blocks at
    ↪   the same time.
43  # Limak is ready to start. You task is to count how many operations will it take him to destroy all towers.
44  \"\"\"
45  destroy_towers(input_str): Returns the number of operations it takes to destroy all towers.
46  parse_input(input_str): Takes a string containing the number of towers on the first line and the heights of the towers on the second and returns
    ↪   the list of heights
47  side_ones(heights_list): From a list of ints, set the first and last elements to 1 and return the list
48  destroy_from_left(side_list): Copy the list and set each each element to the minimum of itself and one more than the element to its left, starting
    ↪   from the second element
49  destroy_from_right(side_list): Copy the list and set each each element to the minimum of itself and one more than the element to its right,
    ↪   starting from the second to last element
50  min_list(l1, l2): Return a list of the minimum of the corresponding elements of l1 and l2
51  to_output_str(min_list): Return the string representation of the maximum element in the list
52  \"\"\"
53  (7 lines)
54
55  # Alex decided to go on a touristic trip over the country.
56  # For simplicity let's assume that the country has $n$ cities and $m$ bidirectional roads connecting them. Alex lives in city $s$ and initially
    ↪   located in it. To compare different cities Alex assigned each city a score $w_i$ which is as high as interesting city seems to Alex.
57  # Alex believes that his trip will be interesting only if he will not use any road twice in a row. That is if Alex came to city $v$ from city $u$, he
    ↪   may choose as the next city in the trip any city connected with $v$ by the road, except for the city $u$.
58  # Your task is to help Alex plan his city in a way that maximizes total score over all cities he visited. Note that for each city its score is
    ↪   counted at most once, even if Alex been there several times during his trip.
59  \"\"\"
60  max_score(input_str): Simple function returning the maximum score Alex can get.
61  parse_input(input_str): Takes a string containing the number of cities and roads on one line, the scores of the cities on the next line, the roads
    ↪   on the next lines besides the last (1-indexed, make 0-indexed), and the starting city on the last line. It returns the city scores, the roads
    ↪   as an edge list, and the starting city.
62  get_neighbors(edges): Returns a dictionary of the neighbors of each city, defaulting to an empty set.
63  get_degrees_and_leaves(neighbors, root): Returns a dictionary of the degrees of each city, and a set of the leaves (excluding the root).
64  remove_leaves(scores, neighbors, degrees, leaves, root): Create a 0-initialized defaultdict of total_extra, and an int of max_extra. Pop leaves
    ↪   until it is empty. Update total_extra and max_extra based on the parent's total_extra vs the leaf's score plus its total_extra, whichever is
    ↪   greater. Return max_extra.
65    pop_leaf(neighbors, degrees, leaves, root): Pop off a leaf. Set parent to sole neighbor of the leaf and delete the leaf from the neighbors
    ↪   dictionary. Decrement the parent's degree. If the parent is not the root and has degree 1, add it to the leaves. Return the leaf and parent.
66  to_output_str(scores, neighbors, root, max_extra): Returns the string of the maximum score Alex can get. If the root isn't in neighbors, return the
    ↪   score of the root. Otherwise, this is the sum of the scores of the cities left in neighbors, plus the returned encountered max_extra.
67  \"\"\"
68  (7 lines)
69
70  # Translate the following solution plan into the above format:
71  {solution_start}{solution_text}
72
73  TRANSLATE to Parsel.
74  \"\"\"
75  """
```

Figure A.47: Translation prompt for APPS programs

# O HumanEval Prompts

We use the same zero-shot prompt to encourage the high-level sketch as in APPS. For translation we use:

```
1  You will aim to solve the following problem in Parsel:
2  {question}
3
4  Translate the following solution plan into Parsel:
5  {solution_start}{solution_text}
6
7  You will translate a solution plan for a problem into Parsel. Each line should contain either
   ↪  a function description or a function reference.
8
9  A function description should be of the form:
10 '''
11 function_name(arg1, arg2): Description of the function
12 '''
13
14 A function reference should be of the form:
15 '''
16 function_name
17 '''
18
19 Use indentation to indicate dependencies between functions. For example, if function A calls
   ↪  function B, then function B should be indented under function A.
20 Make sure that the top-level function matches the name of the function in the solution plan.
```

Figure A.48: Translation prompt for GPT-4

# P   Robotic Plan Evaluation Details

## P.1   Questionnaire

Our questionnaire closely follows that of Huang et al. [28]. We provide a figure with the directions for the accuracy version of the survey in the first image of Fig A.49. We include an example question in the second image. Note that each participant was shown a random 5 questions with their answers in random order. The clarity survey instead asks "For every question below, evaluate how easy it is to understand how the provided steps accomplish the task. Please rank the planned steps for each question from most understandable to least understandable (with 1 as the best and 3 as the worst)." In addition, for the clarity survey, each question text instead said "Rank the following plans based on which is the most understandable (1 = most understandable, 3 = least understandable)."

## P.2   Executability

We find that the automated executability check is a less insightful metric than human evaluation, as it doesn't meaningfully reflect the plan's likelihood of successfully completing a task. Unfortunately, the code to replicate the executability measure from Huang et al. [28] is unavailable. As an alternative, we developed our own executability checker using example code found on VirtualHome's GitHub repository, which evaluates if a proposed plan is syntactically accurate and can be executed within the VirtualHome environment. By leveraging Codex to generate eight Parsel programs for each of the 88 tasks and subsequently compiling them using the Parsel compiler, our method successfully produced executable solutions for all tasks. Conversely, Huang et al. [28] managed executable plans for only 86 tasks. However, it is worth noting that our Parsel compiler explicitly incorporates this executability measure as a constraint, which explains the higher executability rates observed in our approach.

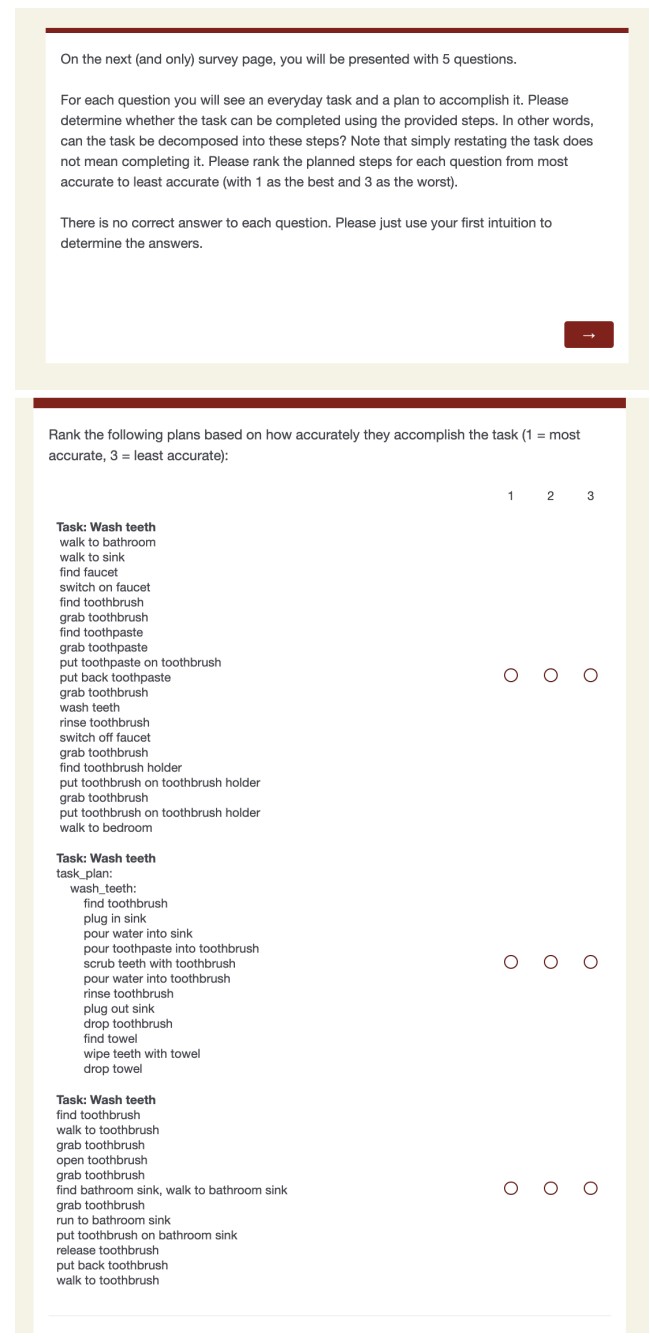

Figure A.49: Screenshot of survey directions and example survey question. In this figure, the first answer was generated by the baseline, the second was the indented Parsel version, and the third was the unindented Parsel version. However, note that the order is randomized for each participant.

## Q Human Case Study

```
1 main(input_string): Takes an input line. First, splits it at newline, and stores the second
   ↪ line in a variable s and the third in t. Splits s in two parts at the asterisk, and
   ↪ checks whether the string t starts with the first part of s and ends with the second
   ↪ part. Returns the string "YES" if that condition is met, otherwise "NO". Also, it should
   ↪  always return "NO" if the length of t is smaller than the sum of the length of the
   ↪ parts of s.
2 "6 10\ncode*s\ncodeforces" -> "YES"
3 "6 10\ncodeforces\ncodeforces" -> "YES"
4 "6 10\ncode*morces\ncodeforces" -> "NO"
5 "6 10\na*a\na" -> "NO"
6 "6 10\na*a\naa" -> "NO"
```

Figure A.50: Solution to https://codeforces.com/problemset/problem/1023/A

```
1 main(input_string): Parses the input and returns the minimum area of the input.
2 "3\n10 1\n20 2\n30 3" -> 180
3 "3\n3 1\n2 2\n4 3" -> 21
4    parse_input(input_string): Takes the input line and first splits on newline. Ignores the
        ↪ first line, and parses each of the remaining lines as a tuple of two numbers, which
        ↪ give a list L of tuples. Returns L.
5    "3\n10 1\n20 2\n30 3" -> [[10, 1], [20, 2], [30, 3]]
6        parse_line(l): Splits l on space, converts each element to int, and returns the result
            ↪  of converting the result to a list.
7        "10 1" -> [10, 1]
8    enumerate_subsets_at_most_k(L, k): Returns all subsets of L with sizes ranging from 0 to
        ↪ k, inclusive.
9    [1, 2, 3], 2 -> [[], [1], [2], [3], [1, 2], [1, 3], [2, 3]]
10       enumerate_subsets(L, k): recusively enumerates the subsets of size k of the list L.
            ↪ Base cases: if k = 0, returns a list containing the empty list. If k > len(L),
            ↪ returns the empty list. Otherwise, first construct the subsets that contain the
            ↪ first element, then those that do not, and return their concatenation.
11       [1, 2, 3], 2 -> [[1, 2], [1, 3], [2, 3]]
12   minimum_area(whs): First, calls enumerate_subsets_at_most_k passing whs and half the
        ↪ length of whs rounded down. Returns the minimum result of calling compute_area on the
        ↪  list given by apply_inversions with whs and the subset.
13   [[10, 1], [20, 2], [30, 3]] -> 180
14   [[3, 1], [2, 2], [4, 3]] -> 21
15       enumerate_subsets_at_most_k
16       compute_area(whs): takes a list of pairs (width, height). Computes the sum of the
            ↪ widths and the maximum of the heights. Returns the product of those two numbers.
17       [[1, 2], [3, 5]] -> 20
18       [[10, 1], [20, 2], [30, 3]] -> 180
19       apply_inversions(whs, subset): Takes a list of pairs of form (w, h) and a subset of
            ↪ indices to invert. Returns a list where the elements of whs whose index is in the
            ↪  subset are inverted to (h, w), and the others appear as given.
20       [[1, 2], [3, 5]], [1] -> [[1, 2], [5, 3]]
21       [[1, 2], [3, 5]], [] -> [[1, 2], [3, 5]]
```

Figure A.51: Solution to https://codeforces.com/problemset/problem/529/B

```
1 main(input): Converts the input to an integer and returns the value of f of n.
2 "1" -> 1
3 "2" -> 3
4 "3" -> 10
5    f(n): First pre-computes the Pascal triangle up to n+1 using compute_pascal_triangle.
     ↪ Then, returns the value of dp(n, pascal_triangle)
6    1 -> 1
7    2 -> 3
8    3 -> 10
9       compute_pascal_triangle(N): returns a matrix with N + 1 rows where m[i][j] corresponds
        ↪  to "i choose k", i.e., the Pascal triangle. It is computed using dynamic
        ↪ programming: m[i][j] = m[i-1][j] + m[i-1][j-1]. All elements are modulo (10**9 +
        ↪ 7). The i-th row has only i columns.
10      2 -> [[1], [1, 1], [1, 2, 1]]
11      3 -> [[1], [1, 1], [1, 2, 1], [1, 3, 3, 1]]
12      dp(n, pascal_triangle): first creates a list with (n + 1) zeros called L. Then fills
        ↪ it in with the following dynamic programming relation: base case is L[0] = 1;
        ↪ then, L[i] = sum (j in [1, i]) pascal_triangle[i-1][j-1] * L[i - j]. Finally,
        ↪ returns the following answer: sum (k in [1, n]) pascal_triangle[n][k] * L[n - k].
        ↪  After each of these assignments, take modulo 10**9 + 7 to avoid big numbers.
13      1, [[1], [1, 1], [1, 2, 1]] -> 1
14      2, [[1], [1, 1], [1, 2, 1]] -> 3
15      3, [[1], [1, 1], [1, 2, 1], [1, 3, 3, 1]] -> 10
```

Figure A.52: Solution to https://codeforces.com/problemset/problem/568/B

```
1 main(input): Reads the input line and counts how many pairs of elements pass the test.
2 "4 2\n2 3\n1 4\n1 4\n2 1" -> 6
3 "8 6\n5 6\n5 7\n5 8\n6 2\n2 1\n7 3\n1 3\n1 4" -> 1
4    parse_input(input): Splits input as a sequence of lines. Each line is parsed as a list of
     ↪  two space-separated integers. The first line of input contains N and P, and the
     ↪ second to last lines are aggregated in a list L. Returns a list with three values: N,
     ↪  P and L.
5    "4 2\n2 3\n1 4\n1 4\n2 1" -> [4, 2, [[2, 3], [1, 4], [1, 4], [2, 1]]]
6    count_valid_pairs(L, p): for each distinct pair (i, j) both ranging from 0 to the length
     ↪ of L, counts how many of those pairs have score at least p in L given by
     ↪ compute_pair_score.
7    [[2, 3], [1, 4], [1, 4], [2, 1]], 2 -> 6
8    [[5, 6], [5, 7], [5, 8], [6, 2], [2, 1], [7, 3], [1, 3], [1, 4]], 6 -> 1
9       compute_pair_score(a, b, L): receives two integers, a and b, and a list of pairs L.
        ↪ Returns how many elements of L contain either a + 1 or b + 1.
10      1, 2, [[2, 3], [1, 4], [1, 4], [2, 1]] -> 2
11      1, 1, [[2, 3], [1, 4], [1, 4], [2, 1]] -> 2
12      0, 1, [[2, 3], [1, 4], [1, 4], [2, 1]] -> 4
13      4, 5, [[2, 3], [1, 4], [1, 4], [2, 1]] -> 0
```

Figure A.53: Solution to https://codeforces.com/problemset/problem/420/C

```
1 main(input): parses the input as two space-separated integers, n and m. Return 2 * f(n, m)
  ↪ modulo 10**9 + 7
2 "2 3" -> 8
3 "3 2" -> 8
4    f(n, m): computes fib(n) + fib(m) - 1
5    2, 3 -> 4
6       fib(m): computes the m-th fibonacci number modulo 10**9 + 7 using dynamic programming
         ↪ starting with dp[0] = 1 and dp[1] = 1, then dp[n] = (dp[n-1] + dp[n-2]) % (10**9 +
         ↪ 7)
7       1 -> 1
8       2 -> 2
9       3 -> 3
10      4 -> 5
11      5 -> 8
12      6 -> 13
```

Figure A.54: Solution to https://codeforces.com/problemset/problem/1239/A

# R Parsel Language Nuances

## R.1 Syntax

**Descriptions.** A function description is represented as a function name followed by comma-separated input arguments in parentheses, and optionally an arrow followed by what the function returns[4], then a colon and text describing the function to be implemented. For example, as part of Conway's Game of Life, one might write

```
count_living_neighbors(grid, i, j): count the number of living neighbors of the cell at the
↪ index (i, j)
```

A function generated from a description can call either the functions defined directly below the description in the function graph (indicated with indentation) or references directly below the description[5], both shown in Fig. ii.

**Constraints.** A constraint is represented as a function's input values comma-separated, optionally followed by an arrow and the expected output of the function. Constraints are provided at the same indentation as the preceding description. For example, after the definition of count_living_neighbors one can write

```
[[1, 0], [0, 1]], 0, 0 -> 1
[[1, 0], [0, 1]], 0, 1 -> 2
```

This indicates that count_living_neighbors should return 1 when called with the arguments `[[1, 0],
[0, 1]], 0, 0` and 2 when called with `[[1, 0], [0, 1]], 0, 1`. Notably, to apply complex constraints on functions, one can describe and constrain higher-order functions. For example:

```
type_fn_output(fn, args): returns the type of the output of a function called with args
count_living_neighbors, ([[1, 0], [0, 1]], 0, 0) -> int
```

This indicates that the function count_living_neighbors should return an integer when called with the input arguments `[[1, 0], [0, 1]], 0, 0`.

What it means to satisfy constraints to validate a program varies from language to language: in Python, one can check that a program passes certain assert statements by evaluating them; however, in a theorem-proving language like Lean, where the ability to run a program (without skipping steps by using `sorry` or `oops` lines) shows that a proof holds, one would instead represent the formal proof statement as the specified constraint (that is, that you are actually proving what you set out to prove). For languages where correctness can be checked without any unit tests, their functions can be treated as also having implicit constraints.

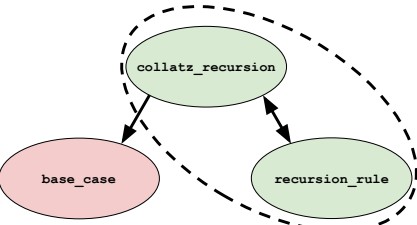

Figure A.55: `collatz_recursion` call graph. There is a strongly connected component formed by `collatz_recursion` and `recursion_rule`.

**References.** A reference is simply the name of a function defined in the current scope (see the next paragraph for details) within the function graph. A reference allows and encourages (via prompt) the parent function to use the referenced function. This allows for recursive function definitions and functions called by multiple functions. For example, one can define an (overly verbose) version of the Collatz conjecture (a well-known recursive open question in mathematics) as shown in Figure ii, where the final line is a reference. We visualize the corresponding call graph and its strongly connected components (SCC) in Figure A.55. In the Collatz functions, `base_case` is implemented first as the `collatz_recursion` SCC depends on it.

---

[4]Note that in Parsel, for Python one can also indicate in the description that a function should yield a value or is a generator.

[5]A nuance here is the optional ability for undefined/out-of-scope functions which are generated by the code LLM to also be implemented automatically.

## R.2 Headers

We also support program headers, allowing global contexts, used when implementing all new functions within a program. This is indicated by a line containing an optional string of special characters (e.g. "#*#*#") separating the body and the text and is passed as a prefix to all prompts.

## R.3 Repeated Automatic Decomposition

As indicated by a rapidly growing number of papers [10, 28], the task of decomposing a task into steps in natural language is one that language models are surprisingly capable of. As explored in concurrent work [57], using language models to automatically and recursively decompose difficult open-ended problems to arbitrary depth is a powerful tool. Thus, we treat the ability to automatically decompose a Parsel function as a key feature of Parsel. This is an optional flag that prompts a language model to generate Parsel code corresponding to any additional subfunctions necessary when Parsel fails to implement a function. These proposed subfunctions are then added as child nodes to the decomposed function node. However, an additional consequence is that Parsel can thus be used to recursively decompose tasks into steps, by repeatedly identifying descriptions that cannot be directly implemented and attempting to decompose them.

## R.4 Scope.

Scope in Parsel is defined by indentation. The scope $S$ of a function $f$ includes the set of functions that can be used as a reference for a given function – that is, all functions where the indentations between the current function to the referenced function are strictly decreasing.

## R.5 Variations due to target language requirements.

Certain aspects of the implementation are still target-language specific. As discussed above, the meaning and representation of a constraint may vary by language. Moreover, every language has a different evaluation function: executing Python is different than compiling and running C++ code, which is different than checking a proof with Lean. Further, every language will likely require a different prompt for the language model. Thus, we detail these particularities in language-specific configuration files.

# S   Pipeline Figure Example

```
 1 Question:
 2 The grand museum has just announced a large exhibit on jewelry from around the world. In the
   ↪ hopes of his potential future prosperity, the world-renowned thief and master criminal
   ↪ Edward Terrenando has decided to attempt the magnum opus of his career in thievery.
 3
 4 Edward is hoping to purloin a large number of jewels from the exhibit at the grand museum.
   ↪ But alas! He must be careful with which jewels to appropriate in order to maximize the
   ↪ total value of jewels stolen.
 5
 6 Edward has $k$ knapsacks of size $1$, $2$, $3$, up to $k$, and would like to know for each
   ↪ the maximum sum of values of jewels that can be stolen. This way he can properly weigh
   ↪ risk vs. reward when choosing how many jewels to steal. A knapsack of size $s$ can hold
   ↪ items if the sum of sizes of those items is less than or equal to $s$. If you can figure
   ↪  out the best total value of jewels for each size of knapsack, you can help Edward pull
   ↪ off the heist of the century!
 7
 8 -----Input-----
 9 Each input will consist of a single test case. Note that your program may be run multiple
   ↪ times on different inputs. The first line of input will consist of two space-separated
   ↪ integers $n$ and $k$, where $n$ ($1 \le n \le 1000000$) is the number of jewels in the
   ↪ exhibit, and $k$ ($1 \le k \le 100000$) is the maximum size of knapsack available to
   ↪ Edward. The next $n$ lines each will describe a jewel. Each line will consist of two
   ↪ space-separated integers $s$ and $v$, where $s$ ($1 \le s \le 300$) is the size of the
   ↪ jewel, and $v$ ($1 \le v \le 10^9$) is its value. Each jewel can only be taken once per
   ↪ knapsack, but each knapsack is an independent problem.
10
11 -----Output-----
12 Output $k$ integers separated by whitespace. The first integer should be the maximum value of
   ↪  jewels that will fit in a knapsack of size $1$. The second should be the maximum value
   ↪ of jewels in a knapsack of size $2$, and so on.
13
14 -----Examples-----
15 Sample Input 1:
16 4 9
17 2 8
18 1 1
19 3 4
20 5 100
21 Sample Output 1:
22 1 8 9 9 100 101 108 109 109
23
24 Sample Input 2:
25 5 7
26 2 2
27 3 8
28 2 7
29 2 4
30 3 8
31 Sample Output 2:
32 0 7 8 11 15 16 19
```

Figure A.56: Pipeline Figure Task

```
1  First, we need to figure out how to calculate the maximum sum of values of jewels for each
   ↪ size of knapsack. To do this, we can use a dynamic programming approach.
2
3  We can create a two-dimensional array of size $n \times k$ where each row represents the
   ↪ different jewels and each column represents the different sizes of knapsack. The value
   ↪ of each cell in the array will represent the maximum sum of values of jewels that can be
   ↪  stolen for a given knapsack size.
4
5  We can then iterate through the array, starting at the beginning. For each row, we can
   ↪ calculate the maximum sum of values of jewels that can be stolen for each knapsack size
   ↪ by taking the maximum of the value of the current cell and the value of the previous
   ↪ cell plus the value of the current jewel.
6
7  At the end, we will have an array that contains the maximum sum of values of jewels that can
   ↪ be stolen for each knapsack size. We can then output these values as our answer.
8
9  This algorithm is correct because we are calculating the maximum sum of values of jewels that
   ↪  can be stolen for each knapsack size by comparing the values of the current cell and
   ↪ the previous cell plus the value of the current jewel. This ensures that we are taking
   ↪ the maximum value for each knapsack size.
```

Figure A.57: Pipeline Figure Sketch

```python
1  # Takes a string containing the number of jewels and knapsack sizes on the first line, the
   ↪ jewels on the next lines, and returns the number of jewels, the knapsack sizes, and the
   ↪ jewels.
2  def parse_input(input_str):
3      lines = input_str.split('\n')
4      first_line = lines[0].split(' ')
5      num_jewels = int(first_line[0])
6      knapsack_sizes = int(first_line[1])
7      jewels = [[0, 0]] * num_jewels
8      for i in range(1, num_jewels + 1):
9          jewel = lines[i].split(' ')
10         jewels[i - 1] = [int(jewel[0]), int(jewel[1])]
11     return num_jewels, knapsack_sizes, jewels
12
13 # Returns the maximum sum of values of jewels that can be stolen for each knapsack size.
14 def max_sum_of_jewels_for_size_and_jewel(jewels, size, jewel):
15     if size < 0:
16         return 0
17     elif jewel < 0:
18         return 0
19     elif jewel == 0:
20         return jewels[jewel][1] if jewels[jewel][0] <= size else 0
21     elif jewels[jewel][0] > size:
22         return max_sum_of_jewels_for_size_and_jewel(jewels, size, jewel-1)
23     else:
24         return max(max_sum_of_jewels_for_size_and_jewel(jewels, size, jewel-1),
25                 max_sum_of_jewels_for_size_and_jewel(jewels, size-jewels[jewel][0], jewel-1)
                     ↪ + jewels[jewel][1])
26
27 # Returns the maximum sum of values of jewels that can be stolen for each knapsack size.
28 def max_sum_of_jewels_for_size(jewels, size):
29     result = []
30     for s in range(1, size + 1):
31         result += [max_sum_of_jewels_for_size_and_jewel(jewels, s, len(jewels) - 1)]
32     return result
33
34 # Returns the string of the maximum sum of values of jewels that can be stolen for each
   ↪ knapsack size.
35 def to_output_str(max_sum_of_jewels_for_size):
36     return " ".join(map(str, max_sum_of_jewels_for_size))
37
38 # Returns the maximum sum of values of jewels that can be stolen for each knapsack size.
39 def max_sum_of_jewels(input_str):
40     '''
41     >>> max_sum_of_jewels('4 9\\n2 8\\n1 1\\n3 4\\n5 100\\n')
42     '1 8 9 9 100 101 108 109 109'
43     >>> max_sum_of_jewels('5 7\\n2 2\\n3 8\\n2 7\\n2 4\\n3 8\\n')
44     '0 7 8 11 15 16 19'
45     '''
46     number_of_jewels, knapsack_sizes, jewels = parse_input(input_str)
47     return to_output_str(max_sum_of_jewels_for_size(jewels, knapsack_sizes))
48
49
50 assert compare_output(max_sum_of_jewels, '4 9\n2 8\n1 1\n3 4\n5 100', '1 8 9 9 100 101 108
   ↪ 109 109')
51 assert compare_output(max_sum_of_jewels, '5 7\n2 2\n3 8\n2 7\n2 4\n3 8', '0 7 8 11 15 16 19')
52 assert compare_output(max_sum_of_jewels, '2 6\n300 1\n300 2', '0 0 0 0 0 0')
```

Figure A.58: Pipeline Figure Sketch

# T  Other Details

## T.1  Test Generation in HumanEval

For our test generation in our HumanEval experiments, we generated a set of tests by prompting GPT-4 to `"Generate an assert-based test for the following function. Answer with only a code block,` ↪ ` and no other text. Do not wrap your asserts in a function.\n" + question` and then collecting and set-aggregating 100 completions.

## T.2  Backtracking

We also support backtracking for the Parsel implementation step, where we re-implement descendants by sampling new solutions for dependencies if a correct solution is not found for a parent. This is necessary to improve the robustness of some of the Appendix examples such as Figure A.14.

## T.3  Training Details

Although we do not train any models, all models used are discussed throughout the paper. See Appendix N for more details about sampling hyperparameters.

## T.4  Error Bars

We estimate a standard deviation of $\pm 1.4\%$ for the best APPS result, given 1000 sampled problems.

## T.5  Compute

The most computationally intensive part of this research, by far (in terms of FLOPS), was the ablation using an open-source CodeGen model, which required several-hundred A100 hours. The rest of the inference was done through API calls.

## T.6  Generated Tokens

For the APPS evaluation, in terms of tokens generated, it is hard to compare the models directly: The CodeT paper does not specify the number of tokens decoded for their evaluation. Without more detail about their evaluations, it is impossible to confidently estimate the tokens generated per program for the CodeT evaluation. The AlphaCode results sample at most 768 tokens per solution, but they do not report average statistics directly - based on Figure 11 in the AlphaCode paper [35], the majority of its generated solutions are 150 to 350 tokens long after removing dead code and comments. The competition-level problems (that we evaluate on) require more tokens on average. For their best reported results, their figures indicate they sample at least 20 billion tokens for the competition-level subset of APPS. On the other hand, for our best results, Parsel generates (on average) 491 tokens of Python code per program implementation, and because we implement each high-level sketch in Python sixteen times (i.e. $k = 16$ in our best $n \times k$), we also sample on average 22 sketch tokens and 43 translation tokens per Python program implementation. Correspondingly, we sample roughly 7 million tokens for our APPS evaluation.

## T.7  Reproducibility

While our contribution is not a model or dataset, we have released our code.

## T.8  Evaluation Time

For a similar API rate limit and evaluation time, we expect our longest-running evaluation runtime would be fairly comparable to the 1,000-program AlphaCode evaluation (despite performing 3x better) and several times faster than the 50,000-program AlphaCode evaluation. In other words, we believe that generating 1,000 programs and evaluating them is not meaningfully faster than generating the 128 Parsel ones and then evaluating their combinations, even in the worst case. Our reasoning is as follows:

*Generation time.* Because AlphaCode has no public-facing API, let us use the advertised rate limit for Codex (40,000 tokens per minute) as the generation speed. (This ignores the request per-minute limit and the number of input tokens in context.) For AlphaCode, assuming roughly 400 tokens per solution, generation would take 10 minutes plus the completion time (i.e., the latency from an API request to the result) of roughly 30 seconds. For Parsel, in our largest-scale experiment, we generate eight plans, their corresponding Parsel programs, and then each Parsel program's corresponding 16 implementations. At the same rate limit and program length, these generations would take about 1.5 minutes, plus three times the completion time. In other words, generation should take about 3 minutes for Parsel vs. 10.5 minutes for AlphaCode.

*Evaluation time.* Let's consider three evaluation scenarios, maintaining the 0.04-second implementation eval timeout used in the paper, run on a single 16-core machine. 1,000 evaluations. If we run 1,000 programs for each method, under the same constraints, their evaluation runtime should be the same. As indicated by Fig. 4-right, the Parsel performance would be substantially higher despite being over 3x faster overall. 10,000 evaluations. If we assume all programs take the maximum allowable runtime (the worst case for Parsel), then 10,000 evaluations takes 6.7 minutes for parsel, compared to 40 seconds for 1,000 AlphaCode program evals. Parsel is still slightly faster, yet solves multiple times more problems. More evaluations. After this point, we run into the 2-minute per Parsel program limit (discussed previously), making Parsel slower than AlphaCode in the worst case — but also solving over 3x as many problems.

*Other nuances.* With all this being said, this is a difficult (potentially impossible) comparison to make fairly for a few reasons. First, AlphaCode is worse than Codex, but likely faster, so it isn't clear which Parsel data point and which rate limit would make an appropriate comparison. Second, runtimes and problem lengths can vary substantially. For instance, we haven't considered input tokens so far, but these penalize AlphaCode more than Parsel, as AlphaCode inferences many more programs. Note that APPS problems are fairly long (the longest competition-level problem requires over 1800 tokens) so the effect of re-processing the problem statement at each program inference is substantial. Since the rate limit includes input tokens and there is a hard limit of 4,000 generated tokens per query, this decreases the effective rate limit by a third!

