# OpenReview forum: "Parsel🐍: Algorithmic Reasoning with Language Models by Composing Decompositions"
_NeurIPS.cc/2023/Conference — NeurIPS 2023 spotlight_

### Official Review · Reviewer_AJpS · 2023-07-03

**Soundness:** 1 poor
**Presentation:** 3 good
**Contribution:** 2 fair
**Rating:** 6
**Confidence:** 4

**Summary:**

The paper proposes Parsel, a framework that decomposes algorithmic reasoning problems into subparts, samples programs for each subpart and verify them. To decompose a problem, Parsel transforms it into an intermediate language that describes the functionality for each subpart and how the subparts depend on each other. To sample programs, Parsel uses existing language models trained on code. To verify the subparts and the whole program, Parsel analyzes their dependencies and use existing test generation techniques.

Parsel is able to improve the performance in solving code problems and virtual robot control.

**Strengths:**

1. The overall idea of divide-and-conquer and verifying subparts is reasonable and intuitive.
2. Using an intermediate language makes both the reasoning and parsing easier.
3. The idea for involving human programmers to help in the loop is inspirational for designing better code assistants.

**Weaknesses:**

### 1. The comparison between Parsel and Codex (codeT improved) is not fair.

Therefore, the claim on L145 that Parsel “substantially improves over prior work, from 14.5% to 25.5%” does not hold.

If I understand correctly, this comparison is demonstrated in Figure 4, where the number 14.5% comes from the blue point labeled **“50 (improved)”** and the number 25.5% comes from the green point labeled **“8x16”**. In my opinion, this comparison is unfair because of two reasons.

- **First, the sample budgets are different.**
    As claimed on L142, the comparison uses an “effective number of complete programs” to represent the sample budget. So the “50 (improved)” setting has a sample budget of 50 while the “8x16” has 128 sample budget. On the own interpolated green curve in Figure 4, Parsel’s performance at sample budget 50 is similar to that of Codex.

- **Second, Parsel uses significantly more evaluation budget.**
    Even if we ignored the first issue and assumed “50 (improved)” and “8x16” had the same sample budget, the comparison would still be unfair because of significant differences in evaluation budgets. “50 (improved)” achieved a pass rate of 14.5% with **50 evaluations**. While ”8x16” achieved 25.5% with **10^6 evaluations**.
    I do understand that one advantage of Parsel is the ability to increase the number of evaluations without more samples, but it’s still unfair to compare the result from 10^6 evaluations with that from 50 evaluations. Because arguably, evaluation budget is more important than sampling budget in code generation, as a failed evaluation would result in penalty in programming competitions and crashes in real production. No one would want a code generation system that gets the correct program after 10^6 failures. It’s both unrealistic and unfair.
    From the reported data, a fairer pass rate for “8x16” would be 4% (when the evaluation budget is 100), which is much worse than “50 (improved)”. To give more idea about how large a 10^6 evaluation budget is and how much improvement it gets, AlphaCode achieves >35% pass rate on CodeContests (Figure 8 from https://arxiv.org/pdf/2203.07814.pdf) with 10^6 evaluations and 6% pass rate with 100 evaluations. Judging from CodeT’s results, CodeContests is harder than competition-level APPS problems.

Because of these two reasons, I find the claim on Parsel’s ability to solve more competition-level problems not reliable.

### 2. Lack of details and analyses on how verification of decomposed functions helps.
Parsel emphasizes the importance of decomposition of the problem into modular functions (components) and the verification of these functions.

While decomposition can be justified by the exponential evaluation budget (though it creates unfair comparison) and the dropping pass rates when the number of chained components is large, how verifying these decomposed functions separately can help is not sufficiently justified, leaving several questions unanswered, to name a few:

- **How are constraints (unit tests) generated for different components generated?** I understand that the test cases for the entire program is generated using CodeT, but how different components are verified is not clear.
- **How many components have unit tests?** The paper claims that for test-less component functions, they can be tested with the test cases for the entire program. I wonder how many components actually have unit tests or if most of them are test-less.
- **Are unit tests mostly correct?** If not, what is the point of using them instead of just using the general test cases?
- **Do testing components separately lead to better perfiormance?**
- **Do components that pass unit tests always build up to a correct program?**
- **How much does partitioning the function reference graph into SCCs help?** The paper claims that for mutually dependent functions, grouping them into SCCs can help save some sampling budget. Does it really?

To me, the results reported look like they are mostly from enumerating different combination of functions and verifying them using global constraints. The claim in your TLDR that the problem is decomposed into subparts and solved by “verifying subparts, and then composing them” does not seem well supported. It’s more like “enumerating the combination of subparts and then verifying them as a whole”.

**Questions:**

I would love to see the answers to the questions in the Weakness section. I may consider raising the score if they are well answered.

**Limitations:**

The authors have addressed the limitations of their work.

---

> ### Author Rebuttal · Authors · 2023-08-10
>
> Thank you for your constructive review and helpful questions!
>
> > As claimed on L142, the comparison uses an “effective number of complete programs” to represent the sample budget. So the “50 (improved)” setting has a sample budget of 50 while the “8x16” has 128 sample budget. On the own interpolated green curve in Figure 4, Parsel’s performance at sample budget 50 is similar to that of Codex.
>
> Overall we agree that Codex 50 (Improved) shouldn’t be directly compared to Parsel 8x16. We believe the scaling properties of Parsel exceed those for Codex, but unfortunately it isn’t possible to verify this with, eg “Codex (Improved) 128”. Codex (Improved) refers to the version of Codex used in [11]. Unfortunately, it’s also hard to extrapolate a comparison here. They do not report more than 50 attempts; however, in their CodeContests results, there is little improvement from 50 to 100 attempts (7.1% to 8.8%). It is, of course, difficult to extrapolate, but a proportional improvement here would represent a change of 3.5% – even with 3x this, it would still underperform Parsel. We attempted to replicate their APPS results with Codex, but could not because 1) they do not provide their few-shot prompt and 2) we found we were rate limited substantially more often than when using Parsel and each generation took longer. (This is likely because we were generating longer completions each time. This made replicating their results at the same scale infeasible, especially given the new restrictions on Codex access.)
>
> > Second, Parsel uses significantly more evaluation budget. … Because arguably, evaluation budget is more important than sampling budget in code generation, as a failed evaluation would result in penalty in programming competitions and crashes in real production… From the reported data, a fairer pass rate for “8x16” would be 4% (when the evaluation budget is 100), which is much worse than “50 (improved)”.
>
> A key motivation behind this paper is the usefulness of tests in generating and identifying useful programs (i.e., test-driven development). Ultimately, whether evaluation or generation is more expensive will depend on use cases, but if the goal is to find a program passing standard unit tests, the compute necessary for generating hundreds of tokens from a large language model is many orders of magnitude more than evaluating an APPS program. There are many such problems where verification is far cheaper than generation. As you mention, given a similar evaluation budget, Parsel performs worse with Codex: this should come as no surprise when it is a format for representing algorithms that Codex has never been trained on. If anything, the fact that Parsel can overcome this gap is noteworthy.
>
> > How are constraints (unit tests) generated for different components generated? I understand that the test cases for the entire program is generated using CodeT, but how different components are verified is not clear… How many components have unit tests? The paper claims that for test-less component functions, they can be tested with the test cases for the entire program.
>
> There are multiple ways to generate constraints. First, one can just generate top-level constraints based on the task, as done in the original CodeT paper. We did this in the HumanEval experiments to make sure that the comparison to CodeT was fair. Second, in the case of human-written Parsel solutions, humans can write tests for relevant components. Third, some of the case studies, like the problem-solving example (Figure A.9), can be solved by generating constraints for all functions and applying the heuristic where we aim to maximize the minimum CodeT score of any function in an implementation.
>
> > Are unit tests mostly correct? If not, what is the point of using them instead of just using the general test cases?
>
> We plan to include more discussion of CodeT [11] in the paper, but one of the key findings was that one doesn’t need all-correct tests to identify correct functions. It’s motivated by the Anna Karenina principle (“All happy families are all alike; each unhappy family is unhappy in its own way”): if many functions pass the same tests, one is likely correct.
> [11]  Codet: Code generation with generated tests. (Chen et al. 222)
>
> > Do components that pass unit tests always build up to a correct program?
>
> Not necessarily - the provided unit tests may not be comprehensive. As mentioned in Appendix S.2, one can also enable a backtracking flag, where a parent component will consider other implementations of child components if a solution cannot be found. However, we did not use this in the large-scale coding experiments, as these relied on unit tests only for the top-level functions.
>
> > How much does partitioning the function reference graph into SCCs help? The paper claims that for mutually dependent functions, grouping them into SCCs can help save some sampling budget… Do testing components separately lead to better perfiormance?
>
> If n functions with m implementations have no mutual dependencies, or if the sizes of the SCCs are no more than c, then partitioning takes it from O(n^m) to O(nm) or O((n^c)/c m). So, for whether it leads to better performance, the answer is certainly: for some Parsel programs like the lisp interpreter (Appendix H), they would be virtually impossible to synthesize without the decomposed testing. For complex programs with many functions, it is infeasible to consider combinations of all of their implementations at the same time.
>
> > The claim in your TLDR that the problem is decomposed into subparts and solved by “verifying subparts, and then composing them” does not seem well supported.
>
> This is a reasonable point. Given that the primary focus of the paper is not on the verification, but rather on the composition and decomposition, we would be more than happy to update the TLDR to “Language models can solve algorithmic reasoning tasks by decomposing them, solving subparts, and composing them.”

---

> > ### Comment · Reviewer_AJpS · 2023-08-10
> >
> > Thank you for the response! Most of my concerns are well addressed.
> >
> > However, I'm still not convinced by your argument on generation budget vs evaluation budget. I understand that in terms of computation cost, running large language models is way more expensive than running the programs they created. I think our disagreement might be on the meaning of tests here. It seems to me that you treat system tests (in codecontests) as merely a metric for correctness that language models and humans have access to during development. But to me, failing tests in a coding contest is the same as crashing in a production environment, which leads to serious consequences. It's not the computation cost of evaluation I'm talking about, it's the risk and consequences of program failure. Therefore, I'm not against using a large evaluation budget on your generated unit tests, but I am against an unfairly large evaluation budget on the system tests.

---

> > > ### Author Response · Authors · 2023-08-11
> > >
> > > There's a genuinely interesting philosophical question here. The question we (and, in our view, most prior work) were looking at with APPS is, given these particularly hard problems, limited by your generation budget, what's the chance you find a correct solution? But if you instead view each test as a successive submission, where any failure is like a production crash, you would almost certainly take a different approach in practice -- one more like the one we took in the HumanEval test generation experiments, one that uses interpreter feedback to revise solutions, or perhaps even one that uses the language model to simulate the result of tests. These two interpretations are important, but they highlight different concerns: if you can guarantee 100% success on the first submission on a problem, but it requires a trillion generated tokens, that's also not very useful in practice. A method's ability to efficiently generate reasonable solutions in the first place is one key bottleneck, and assuming an imperfect model, the ability to identify them is another. We believe a discussion around this would improve the paper, so we intend to add this - thank you! We could also run a pass@1 evaluation using generated tests, like in the current HumanEval experiment, for the camera-ready.
> > >
> > > Lastly, if you believe we've addressed most of your concerns, we would sincerely appreciate it if you raised your score.

---

> > > > ### Comment · Reviewer_AJpS · 2023-08-11
> > > >
> > > > Thank you!
> > > >
> > > > I do believe this trade-off between sample budget and evaluation (or system failure) budget should be considered. And I would appreciate it if you discussed this issue and made it clear in the abstract that parsel is using a much larger evaluation budget in exchange for smaller sample budget, because I find the current figures (especially figure 4 Left) a bit confusing for those who could be assuming that sample budget is on a similar scale to evaluation budget.
> > > >
> > > > Since my other concerns are addressed and you promised to discuss this issue, I'm raising my score to 6.

---

### Official Review · Reviewer_9Jji · 2023-07-04

**Soundness:** 3 good
**Presentation:** 3 good
**Contribution:** 4 excellent
**Rating:** 7
**Confidence:** 4

**Summary:**

The paper presents a system called Parsel for program synthesis. First, a LLM predicts a Parsel program given a task specification or natural language plan. The Parsel program contains a hierarchy of functions which themselves might have functions inside. Each function is described with a function signature, a natural language description of what it does, and sometimes constraints (e.g., I/O examples) and references to other functions to call. Then, the Parsel synthesizer attempts to translate the Parsel program into a traditional program, e.g., a Python program. The LLM generates multiple implementations of each function and a combinatorial search is performed to find combinations of implementations that lead to passing the constraints. Overall, the Parsel approach is shown to outperform directly generating solutions with Codex and AlphaCode on the APPS dataset of competitive programming problems, when controlling for the number of complete programs generated. Other experiments are performed on HumanEval (simple Python programming problems) and VirtualHome (robotic planning).

**Strengths:**

Originality: The Parsel language and overall approach is quite novel and creative, and in my opinion this is the paper’s biggest strength. Parsel’s motivation is clear: we want to decompose problems into multiple functions or groups of functions, such that each group can be implemented independently. However, it was previously not clear how this goal should be achieved, as it requires a planning step that decides how the problem should be decomposed. The Parsel approach nicely resolves this by providing a natural outline of the hierarchical decomposition, containing just the right amount of information to facilitate synthesis of individual functions, while allowing different implementations to be explored.

Quality: The Parsel approach makes a lot of sense from a technical perspective. The experiments explore various aspects of Parsel.

Clarity: The motivation, intuitions, and related work are well written.

Significance: These results are quite significant themselves, and the problem being tackled (program synthesis) is important too.

**Weaknesses:**

In my opinion, the paper’s main weakness is the lack of a deep analysis of *why* Parsel has its good performance. The approach has many parts and it is unclear which parts are the really important ones. Here are some potential reasons why Parsel is good, and potential experiments to gauge the importance of those reasons:

1. Predicting a **high-level sketch or plan** helps the model gain an overall understanding of the problem in a chain-of-thought kind of way, leading to better downstream predictions.

    * Reason 1 is partially analyzed in Lines 156 - 164 (when Codex is given the high-level plan, it performs much worse than Parsel given that plan). But I’m also interested in the other kind of ablation: how well does Parsel do without the high-level plan, going straight from the task specification to the Parsel program (on the APPS dataset)?

2. The **Parsel program itself** makes synthesis easier, since it contains a good decomposition with just the right amount of information for easy synthesis of the individual functions.

    * What if we replaced the Parsel program with a Codex prediction of the entire solution, which we then use as an outline? Specifically, we prompt Codex to implement a code solution that is decomposed into helper functions with docstrings. (If a function is not generated with a docstring, we can prompt an LLM to predict a docstring given the function and other relevant context from the task.) Then we take the function signatures, docstrings, and references (call graph) from the solution to serve as a replacement for the Parsel program. Thus, we sample new implementations of the individual functions and search for implementation combinations as usual.

3. The approach enables **trying many combinations of implementations** with a low cost (effective number of complete programs sampled).

    * What if we disallowed trying multiple combinations of implementations, i.e., only used pass@n x 1?

4. The authors might identify other potential reasons that I didn’t think of.

By including these sorts of ablations, we would get a much more clear understanding of the relative importance between these factors. This could help readers gain a deeper intuition behind why Parsel works, so they may better adapt or extend the approach in future work. Even though my rating is positive right now, I *strongly encourage* the authors to perform these ablations, because I believe the paper will be much stronger (an amazing paper) with a deeper analysis of Parsel’s core ideas.

Another weakness of the paper is that the writing could be made clearer about the algorithm and experiments. Please see the questions below.

Please note: I will be happy to raise my score if I think these weaknesses are adequately addressed in a revision.

**Questions:**

## About the algorithm

If there is a function with constraints where none of the k implementations of the function pass the constraints, then what happens? I assume this causes the entire Parsel program to be unsuccessful, but this is not explicitly stated in the text. Does this lead to a tradeoff where, if the decomposition is too detailed with many individual functions, then there is a higher likelihood of a single function having no correct implementation? And, intuitively it seems useful to include constraints for all functions where the LLM can predict constraints confidently, but then more functions with constraints also leads to a higher likelihood of a single function having the wrong constraints. Can you comment on this “weakest link” issue?

There is an exponential blowup in the number of function implementation combinations. I understand that SCCs of the call graph and test dependency graph are used to reduce the number of functions considered at once, but still the exponential blowup remains. Do you impose some upper limit on the amount of combinations tried? Line 113 mentions *samples* -- when do you sample versus try all combinations, and how many samples are used?

Does a Parsel program always define a single outermost function that serves as the solution to the problem specification? This would make sense given the rest of the algorithm but is not explicitly stated.

Once we find an implementation for a function (or SCC) passing its constraints, is the implementation “locked in” permanently or does the algorithm also consider other implementations/combinations later? In other words, do we end up with at most 1 complete implementation for each Parsel program, which is then checked against the problem specification (possibly with hidden test cases)? Or, might we continue to search for other implementations of the Parsel program and evaluate those on the hidden tests too?

The CodeT score is a core part of the approach in the case of generated tests and is referenced repeatedly for the HumanEval experiments. The paper should provide some background on CodeT.

I know that Appendix F contains Parsel pseudocode in Parsel style, which might clear up some of the questions above. I did not read that pseudocode (Figure A.11 and A.12) carefully because it’s a full page long, and I hope the authors can write a more succinct version in a more traditional style in the main paper. I also think Parsel programs are not as clear as actual pseudocode, because they lack detail about how inner functions are used to implement the outer function. It’s as if one had to understand how a Python program worked, but only seeing the function signatures, docstrings, and call graph... many important details would be lost.

## About the experiments

What I/O examples are given in the problem specifications, and are there held out tests used to evaluate the correctness of synthesized programs? The description of the experiment setup is lacking these kinds of details.

Is it expected that the LLM can predict *correct* constraints in terms of I/O examples for individual functions? It is commonly observed that LLMs are quite bad at executing code or even performing arithmetic. I would expect that many constraints would have subtle errors that lead to incorrect implementations being selected from the search over function combinations. For example, is the constraint in Figure 2 (ii) directly copied from the problem specification, or does the LLM generate it from scratch? Can multiple constraints be added to a single function?

Line 150 mentions that using the LLM to generate programs is much more computationally expensive than running programs on a CPU. This is certainly true in general. However, it is unstated how many programs are actually run (considering the exponential number of implementation combinations being considered), and Figure 4 appears to have a datapoint close to 1 million program evaluations per problem. Because APPS is a dataset of competitive programming problems, I’d expect many programs to have long-running evaluations, either from implementation errors (infinite loops) or suboptimal algorithmic design. Even if only 1% of programs run until timeout, the time limit is set to 1 second (a small time limit in actual competitive programming competitions), and we assume that all other programs evaluate instantly, then 1 million programs would evaluate for 2.8 CPU-hours, per problem. Such an amount of compute cannot be swept under the rug and should be discussed! Certainly from an end-user perspective, waiting several seconds for an API call to GPT-4 feels much better than waiting for multiple CPU-hours of program executions. Appendix E.3 does mention parallelizing program executions but the resulting wallclock time per problem is not mentioned.

For the HumanEval experiment, the writing is less clear about what the main conclusion or takeaway should be. For example, could you summarize Lines 175 - 194 in 1 sentence? The writing would be improved by adding that sentence, and cutting back on some of the details. Similarly, Lines 199 - 204 contain many details that are not useful for readers who do not dig up Parsel’s solutions to the problems mentioned.

For the VirtualHome experiment, it is not clear how to interpret the numerical results. Does Figure 7 say that the Codex (baseline) plans were the *most preferred among the 3 plans* roughly 30% - 35% of time, which is actually not bad for the baseline? How exactly does the comparison work, considering that Parsel has 2 attempts but the baseline only has 1? Does “X is 70% more likely to be preferred over Y” (Line 283) imply P(X preferred) = 85% and P(Y preferred) = 15% such that 85% - 15% = 70%, or something else? Why not just report P(X preferred) directly? What exactly does “X preferred” mean: “one of the 2 Parsel programs was better than the baseline”, or “the indented Parsel program was better than the baseline”?

Overall I am less excited about the VirtualHome experiments because of the lack of rigorous correctness checking (instead relying only on rankings of accuracy and clarity). The example in Figure A.49 does not alleviate my concerns as the Parsel solutions seem much worse than the baseline’s solution, which itself is not completely correct either. What is the main takeaway from the VirtualHome experiments that was not evident from the other experiments?

I know it is hard to investigate contamination issues considering the opacity of some LLMs, but the potential issue should be discussed in greater detail. For example, is it possible that GPT-3 has seen solutions to the APPS problems while AlphaCode and Codex have not? For the Lisp example (Line 354), it seems likely that the code LLM has already seen example implementations of the Lisp interpreter, and it recognizes the general patterns even without any mention of the word “Lisp” in the Parsel program.


## Minor questions and suggestions
* Line 99: “Parsel implements functions with **a** post-order traversal from **the** function at the root”
* Figure 4 caption: a period is missing after “AlphaCode [35]”
* Figure 4 caption: “we examine to understand the effect of evaluation number” is awkward phrasing
* Line 165: missing space between “HumanEval” and “[12]”
* Line 319: missing some punctuation (comma or period) between “model” and “and”
* Line 347: missing space between $\ge$ and 0.99

**Limitations:**

One undiscussed limitation is that Parsel likely requires more expensive LLM calls for the same effective number of complete programs sampled, for three reasons:

1. Each transformation from task specification to the high-level plan and then to the Parsel program requires an LLM call, accounting for many tokens that would not be needed by the Codex baseline.
2. Each function is implemented separately which requires encoding a different prompt for each function with function-specific context.
3. The decomposition of the entire solution into many functions might lead to more total code tokens sampled compared to a solution obtained without explicit decomposition.

The paper could use some discussion about these factors, potentially summing the total LLM cost per problem for Parsel versus Codex.

---

> ### Author Rebuttal · Authors · 2023-08-10
>
> Thank you so much for the thorough and helpful review. We sincerely appreciate the attention to detail and the encouraging and supportive framing. Note we are constrained to 6,000 characters here (a new rule…), but are happy to elaborate during discussion.
> > I’m also interested in the other kind of ablation: how well does Parsel do without the high-level plan?
>
> Great suggestion! On a 200-problem competition-level APPS sample we found that, without a high-level plan, the accuracy falls to 13% (from 25.5%, w/ the same configuration). This is better than the ablation where Parsel wasn’t used, suggesting that Parsel plays a larger role than the high-level plan on APPS but both are necessary. Note part of this improvement is from the challenge of prompting Codex to generate Parsel, given long APPS problem statements. We’ll include the new prompt in the Appendix.
> > What if we replaced the Parsel program with a Codex prediction of the entire solution?
>
> This is an insightful proposal; we explored something very related. There are some challenges that initially limit its generality, many of which are solvable but collectively likely an entirely new work. The main ones are 1) extracting a call tree across many kinds of Python solutions and 2) getting Codex to generate meaningful standalone function descriptions. This may be possible by backtranslating generated Python solutions to Parsel (which does not require Codex to know anything about Parsel) – see Appendix K – but for these reasons there are many functions where that doesn’t work.
> > What if we disallowed trying multiple combinations[?]
>
> We touched on this in Fig. 8, but it should be much more clearly signposted. We interpret the results to suggest that the number of Parsel programs and the number of combined implementations play similar important roles. In other words, one can compensate for implementation ability with Parsel-generation ability and vice-versa (to a limit).
> > [What if] there is a function with constraints where none of the k implementations of the function pass? … Once we find an implementation … is [it] “locked in”?
>
> There are a few things that happen, depending on flags and context. By default, if no implementation passes the provided constraints, this is a failure. But, we also support backtracking (see Appendix S.2) - if enabled, if a parent fails to pass its constraints, it can reattempt its children - otherwise child implementations are locked in. When generating all tests and using CodeT scoring, then aside from some naive heuristics (e.g. a function with no passed constraints), detecting failure is difficult.
> > Is it expected that the LLM can predict correct constraints in terms of I/O examples for individual functions?
>
> Not necessarily – CodeT found that one doesn’t need all-correct tests to identify correct functions. It’s motivated by the Anna Karenina principle (“All happy families are all alike; each unhappy family is unhappy in its own way”): if many functions pass the same tests, one is likely correct.
> > Line 150 mentions that using the LLM to generate programs is much more computationally expensive than running programs... Do you impose some upper limit on the amount of combinations tried?
>
> Good point! We sample and test up to 100,000 items with a timeout of 0.04 seconds each, and also have a per-problem two minute limit – noted in Appendix M but we’ll highlight it. Note, if we’d used LLM-level compute per problem, this evaluation would’ve been near-instant (also easier to speed up evaluations via parallelization vs LM generation).
> > For the HumanEval experiment, the writing is less clear about the takeaway… Could you summarize in 1 sentence?
>
> Excellent point; “Given the same set of generated tests and program generation budget, and selecting only one best solution, Parsel significantly increases the probability that it’s correct.”
> > For the VirtualHome experiment, it is not clear how to interpret the numerical results.
>
> You’re right, this is confusing and we’ll clarify. Each comparison is pairwise (not best of three). Figure 7 compares only the (non-hierarchical) Parsel-generated plan and the Codex-generated solution. In other words, “the non-indented Parsel program was better than the baseline” (and note there was no preference on indentation). I.e. when people compared non-indented Parsel to the baseline, they said it was more accurate two thirds of the time.
> > What is the main takeaway from the [VH] experiments[?]
>
> There are two: First, they show Parsel's flexibility in handling more open-ended tasks beyond traditional code generation on an increasingly relevant LM reasoning domain. Second, they offer a chance to evaluate if the Parsel-generated plans are more clear – the results suggest they are. However, we strongly agree more robust metrics are needed in this emerging domain of LLM-based robotic planning.
> > [Is] it possible [GPT-3] has seen solutions to the APPS problems while AlphaCode and Codex have not? [T]he code LLM has already seen examples of the Lisp interpreter
>
> They’re based on one model so, while it’s sadly hard to know for sure, it’d be a bit surprising if code data was used to train the text version but not the code version. As for lisp, good point – we’ll add this. But note, many descriptions are not obviously Lisp-related.
>
> 1. *Does a Parsel program always define a single outermost function?* Yep!
> 2. *The paper should provide some background on CodeT.* We'll be happy to elaborate in the related works.
> 3. *I hope the authors can write [the pseudocode in a] traditional style.* We’ll revise Figure A.11 - also allowing a useful comparison of Parsel to traditional pseudocode.
> 4. *What I/O examples are given?* For APPS and HumanEval, there are public tests in problem statements and private tests for evaluation.
> 5. *Parsel likely requires more expensive LLM calls.* It depends! It’s discussed in App. S.6 but we agree that highlighting this clearly and discussing these costs more would be good.

---

> > ### Comment · Reviewer_9Jji · 2023-08-16
> >
> > Thanks for the clarifications and new ablation.
> >
> > > trying many combinations of implementations
> >
> > Figure 8 helps a bit, but I'm curious for more specifics. You might say something like:
> >
> > "With a sample budget of 50 programs, Codex (Improved) solves 14.5% of APPS problems. Instead, we could sample 5 Parsel programs with 10 implementations each. If we directly test these 50 implementations without mixing-and-matching functions, we would solve only **X%** of problems. However, after mixing-and-matching functions (trying **Y** combinations on average), the pass rate significantly improves to **Z%**."
> >
> > Is it easy to fill in those numbers? With this comparison (or a similar one), the paper can better quantify the benefit provided by mixing and matching functions.
> >
> > > We sample and test up to 100,000 items with a timeout of 0.04 seconds each, and also have a per-problem two minute limit – noted in Appendix M but we’ll highlight it.
> >
> > I'm confused about the 100K items. The highest point in Figure 4-left is the 25.5% highlighted on line 145, pass@8x16. If you claim this result is with only 100K items, then what is the highest point on Figure 4-right, which also looks like 25.5%, pass@8x16, but looks close to 1 million items on the log-scale x-axis? The point on Figure 4-right at 100K items looks around 22%, definitely not >25%.
> >
> > 2 minutes per problem is a reasonable CPU compute budget. Please do highlight this -- otherwise, it is easy for readers to dismiss the paper by thinking "they just swap implementations of helper functions, and _of course_ you'd get good results after spending (literally) hours trying a million different combinations".

---

> > > ### Author Response · Authors · 2023-08-17
> > >
> > > Thank you very much for the follow-up and the suggestions!
> > >
> > > > Figure 8 helps a bit, but I'm curious for more specifics.
> > >
> > > Got it; we've now run the suggested ablation. It is somewhat similar to the one described in lines 156-164, but instead of skipping the Parsel step, we generate the implementations based on the Parsel program (and consider entirely independently generated implementations), and consider 48 implementations. To maximize the use of cached generations, we evaluated 6 Parsel programs and 8 implementations each (48 implementations per problem). We believe the most salient comparison to be the following: with these 48 Parsel-generated implementations, the performance was only 3%; on the other hand, when combining them, the overall pass rate with 6 Parsel programs and 8 implementations was 14.6%, with the only difference being that we considered combinations (specifically, 34k combination evaluations on average per Parsel program, with a limit of 100,000 evaluations per Parsel program).
> > >
> > > Note that 3% is from a sample of 200 random competition-level problems, 34k is from a sample of 200 random Parsel programs, and 14.6% is across all competition-level problems. We also want to highlight that we believe the scaling properties of Parsel exceed those for Codex, but unfortunately, it isn't possible to verify this directly with, e.g., "Codex (Improved) 128". Unfortunately, they do not report more than 50 attempts; however, in their CodeContests results, there is little improvement from 50 to 100 attempts (7.1% to 8.8%). We attempted to replicate their APPS results with Codex but could not because 1) they do not provide their few-shot prompt, and 2) we found we were rate limited substantially more often than when using Parsel and each generation took longer.
> > >
> > > > I'm confused about the 100K items.
> > >
> > > We sincerely apologize for this miscommunication: we should have specified that that's 100,000 attempts per Parsel program (which is why that rightmost point is 800,000, corresponding to the 8x16 point) and that the 2-minute per-problem limit is also per Parsel program. Otherwise, a Parsel program with only ten combinations might never be run because the others have many more combinations.
> > >
> > > Thank you again for the questions - we hope these results help paint a clearer picture of Parsel's strengths and limitations! We'll also incorporate these details into the paper.

---

> > > > ### Comment · Reviewer_9Jji · 2023-08-17
> > > >
> > > > Thanks for the clarifications.
> > > >
> > > > Considering that Parsel (in the 8x16 setting used for the numbers referenced in the paper) spends up to 16 minutes trying combinations of functions, which is definitely not negligible from a usability perspective (even though the CPU cost is smaller than LLM inference cost), it would be very helpful to include commentary on elapsed time.
> > > >
> > > > Imagine Parsel and AlphaCode were run on a normal modern desktop computer, with multiple CPU cores but no access to any CPU cluster, querying public LLM APIs with their normal rate limits. After making some reasonable assumptions (number of cores, time limit per execution, LLM rate limit), can we break down how long each method would take?
> > > >
> > > > My initial feeling is that Parsel would take perhaps 2-5x as long as AlphaCode, due to needing extra prompts (creating Parsel programs) and program execution. If this is the case, then this is a limitation of Parsel that should be mentioned in the paper.

---

> > > > > ### Author Response · Authors · 2023-08-18
> > > > >
> > > > > Agreed, the evaluation time is worth being explicit about! For a similar API rate limit and evaluation time, we expect our longest-running evaluation runtime would still be fairly comparable to the 1,000-program AlphaCode evaluation (despite performing 3x better), and several times faster than the 50,000-program AlphaCode evaluation. In other words, we believe that generating 1,000 programs and evaluating them is not meaningfully faster than generating the 128 Parsel ones and then evaluating their combinations, even in the worst case. Let us explain this estimate:
> > > > >
> > > > > *Generation time.* Because AlphaCode has no public-facing API, let us use the advertised rate limit for Codex (40,000 tokens per minute) as the generation speed. (This ignores the request per-minute limit and the number of input tokens in context.) For AlphaCode, assuming roughly 400 tokens per solution, generation would take 10 minutes plus the completion time (i.e., the latency from an API request to the result) of roughly 30 seconds. For Parsel in our largest-scale experiment we generate eight plans, their corresponding Parsel programs, and then each Parsel program's corresponding 16 implementations. At the same rate limit and program length, these generations would take about 1.5 minutes, plus three times the completion time. In other words, generation should take about 3 minutes for Parsel vs. 10.5 minutes for AlphaCode.
> > > > >
> > > > > *Evaluation time.* Let’s consider three evaluation scenarios, maintaining the 0.04-second implementation eval timeout used in the paper, run on a single 16-core machine. **1,000 evaluations.** If we run 1,000 programs for each method, under the same constraints, their evaluation runtime should be the same. As indicated by Fig. 4-right, the Parsel performance would be substantially higher despite being over 3x faster overall. **10,000 evaluations.** If we assume all programs take the maximum allowable runtime (the worst case for Parsel), then 10,000 evaluations takes 6.7 minutes for parsel, compared to 40 seconds for 1,000 AlphaCode program evals. Parsel is still slightly faster, yet solves multiple times more problems. **More evaluations.** After this point, we run into the 2-minute per Parsel program limit (discussed previously), making Parsel slower than AlphaCode in the worst case — but also solving over 3x as many problems.
> > > > >
> > > > > *Other nuances.* With all this being said, this is a difficult (potentially impossible) comparison to make fairly for a few reasons. First, AlphaCode is worse than Codex, but likely faster, so it isn't clear which Parsel data point and which rate limit would make an appropriate comparison. Second, runtimes and problem lengths can vary substantially. For instance, we haven't considered input tokens so far, but these penalize AlphaCode more than Parsel, as AlphaCode inferences many more programs. Note that APPS problems are fairly long (the longest competition-level problem requires over 1800 tokens) so the effect of re-processing the problem statement at each program inference is substantial. Since the rate limit includes input tokens and there is a hard limit of 4,000 generated tokens per query, this decreases the effective rate limit by a third!

---

> > > > > > ### Comment · Reviewer_9Jji · 2023-08-21
> > > > > >
> > > > > > Thank you for the clarifications and extra details. Due to the extra ablations and commentary provided by the authors, the weaknesses in my review have mostly been addressed.
> > > > > >
> > > > > > I only have one suggestion remaining -- I still think "What if we replaced the Parsel program with a Codex prediction of the entire solution?" would be a very interesting experiment. This would be a compelling baseline motivated by the following thought process: We want to spend more CPU time on evaluating programs with the same LLM sample budget, so we will re-sample helper functions and try different combinations of their implementations. What is the most straightforward way of achieving this? Let's encourage the LLM to generate a solution decomposed into multiple functions. Then, we'll sample alternative implementations of individual functions, but to do so, we also need a description of what the function should do (a docstring), and what other functions it should call (from the original solution's call graph). Finally we can mix and match implementations just like Parsel does, but without a need for the Parsel language.
> > > > > >
> > > > > > Regardless, I think the authors have done enough for this paper already. (Of course, the authors should still make sure to revise the paper's text to clarify any parts that were unclear or discussed during the review period.)
> > > > > >
> > > > > > I am raising my score from 6 to 7.

---

> > > > > > > ### Author Response · Authors · 2023-08-21
> > > > > > >
> > > > > > > We also think that would be a fascinating experiment - however, there are enough practical challenges to implementing the "Codex solution -> sets of implementations" (especially, 1. you need docstrings that stand on their own, aside from their immediate children, and 2. automatically getting call graphs across the many kinds of generated Python programs is much harder than one might expect, but also many other nuances) that we think it could be an entire additional paper.
> > > > > > >
> > > > > > > Thank you again for the valuable discussion and insights! We sincerely believe they've improved the paper.

---

### Official Review · Reviewer_fDT5 · 2023-07-06

**Soundness:** 3 good
**Presentation:** 3 good
**Contribution:** 4 excellent
**Rating:** 7
**Confidence:** 5

**Summary:**

This paper proposes Parsel, a framework for algorithmic reasoning with LLMs. Parsel can be seen as a kind of “programming language” that is implemented with mostly natural language which describes the functionality of the program -- that is, the algorithmic reasoning plan. Then, the Parsel synthesizer can translate the plans into corresponding languages like Python. Finally, the reasoning problem can be solved by executing the synthesized Python code. All the processes are completed by analyzing the Parsel program and prompting LLMs. Experiments show that Parsel can improve the LLM’s coding ability and generate preferred robotic plans.

**Strengths:**

1. Parsel reveals the power of hierarchical planning and decomposition for complex reasoning tasks.
2. Parsel is a step towards a new kind of programming paradigm. That is, people can program more naturally with Parsel which lowers the programming barrier.
3. Parsel might be a new kind of prompting method for complex reasoning tasks. Users (or LLM) can write a Parsel program first and then the Parsel synthesizer can automatically decompose the Parsel program into small function pieces and prompt LLM with these pieces, and the synthesizer should compose these pieces to form a whole function.
4. The result on code generation is promising, although I still have some questions about it.

**Weaknesses:**

1. Although Parsel can be generated by LLMs, the Parsel synthesizer is not reliable for human users. When coding with Parsel and the synthesized Python code is wrong, users do not know whether the LLM for synthesizer is not working correctly, or if there are bugs in the Parsel code they wrote themselves.
2. Also, there are no debugging operations provided in Parsel. Users cannot locate the bugs in Parsel, and they cannot debug on Parsel.
3. The correctness of Parsel synthesizer relies on IO constraints heavily: It seems that IO constraints are the only guarantee for the correctness generation. This is a little bit weird because the main part of Parsel is the natural language description, which is where users spend a lot of time. However, to debug Parsel and ensure the final result is correct, a more effective approach is to provide more detailed and fine-grained IO or allow the LLM to just regenerate to meet IO, rather than making modifications to the natural language or code structure.
4. The main experiment on APPS (figure 4) doesn’t seem like a fair comparison: Parsel is provided with detailed plans generated with GPT-3 while baselines are not. Providing results like “plans generated by codex (if possible)” “baseline results on GPT-3/4” or just more results with different budget settings on the “ablating the Parsel synthesizer” part would be good.
5. Experiment settings are not clear, see questions.

**Questions:**

1. Baseline settings are not well explained in figure 4: What does “codex improved” mean? The improvement relative to this baseline is very marginal.
2. Does Parsel use test generation in the evaluation? Do baselines use it?
3. Figure 1 shows Parsels ability in multiple languages, but the main paper only showcased Python, making figure 1 overstated.
4. The github link is 404.

**Limitations:**

Limitations are discussed well in the paper.

---

> ### Author Rebuttal · Authors · 2023-08-10
>
> Thank you for your insightful comments – the use of Parsel by human users was a key motivation, so these questions are really useful!
>
> > Although Parsel can be generated by LLMs, the Parsel synthesizer is not reliable for human users. When coding with Parsel and the synthesized Python code is wrong, users do not know whether the LLM for synthesizer is not working correctly, or if there are bugs in the Parsel code they wrote themselves… Also, there are no debugging operations provided in Parsel. Users cannot locate the bugs in Parsel, and they cannot debug on Parsel.
>
> We absolutely recognize and acknowledge that there are changes that could be made to improve the usability for human users. Indeed, this is a direction we are actively pursuing. However, there are some useful debugging features already present. For example, because users can write constraints/unit tests, and the program is implemented from the leaves up, it is often clear which parts of the program are failing and which unit tests they are failing on. In other words, it is often unambiguous when functions are somehow underspecified, whether in language or in constraints. In addition, due to caching, errors are not generally stochastic.
>
> > The main experiment on APPS (figure 4) doesn’t seem like a fair comparison: Parsel is provided with detailed plans generated with GPT-3 while baselines are not. Providing results like “plans generated by codex (if possible)” “baseline results on GPT-3/4” or just more results with different budget settings on the “ablating the Parsel synthesizer” part would be good.
>
> Thank you for this point! There are two things worth noting here (and also worth highlighting in the paper). First, the GPT-3 model we used and Codex are both based on the same model, which makes it somewhat less likely that any improvements come from GPT-3 simply being better (though clearly not a guarantee). Second, since the ablation highlighted in the paper clearly indicates that the intermediate plan is not sufficient for the observed performance improvement, we felt it would also be valuable to test whether it was necessary. We thus conducted an additional experiment where we skipped the plan generation step and instead asked Codex to generate Parsel directly. We observed that on a sample of 200 random competition-level APPS problems, generating Parsel directly from the problem solved 13% of them. Given that this is a more substantial improvement than the Parsel ablation, this suggests that, on this dataset, Parsel plays a larger role than the high-level plan, but both are necessary. Note that we used three few-shot Parsel examples for this experiment, as we observed that including many few-shot Parsel translation examples would result in it disregarding the problem statement, which for APPS is often quite long (e.g., a page of plaintext).
>
> > Baseline settings are not well explained in figure 4: What does “codex improved” mean? The improvement relative to this baseline is very marginal.
>
> Codex (Improved) refers to the version of Codex used in [11]. Unfortunately, it’s hard to make a direct comparison here - from 10 to 50 solutions, their performance improves from 6.3% to 14.5%. They do not report more than 50 attempts; however, in their CodeContests results, there is little improvement from 50 to 100 attempts (7.1% to 8.8%). It is, of course, difficult to extrapolate, but a corresponding improvement here would only represent a change of 3.5% – even with 3x this improvement, this would still underperform our result. We attempted to replicate their APPS results with Codex, but could not because 1) they do not provide their few-shot prompt and 2) we found we were rate limited substantially more often than when using Parsel and each generation took longer. This is likely because we were generating longer completions each time. This made replicating their results at the same scale infeasible, especially after we were unable to split up evaluation between members of the team following the new restrictions on Codex.
>
> > Does Parsel use test generation in the evaluation? Do baselines use it?
>
> In the HumanEval comparison, we include test generation when we comparing to CodeT pass@1, using the same generated tests for both. In the APPS eval, we do not include test generation.
>
> > Figure 1 shows Parsels ability in multiple languages, but the main paper only showcased Python, making figure 1 overstated.
>
> The two examples in Figure 1 correspond to the programming and robotic planning sections of the paper. In addition, we include examples of Parsel generating Lean (formal theorem proving) in the appendix.
>
> > The github link is 404.
>
> We anonymized the GitHub URL to maintain anonymity. However, we released our code (indeed, we already have, and it is actively being used by others outside our team, with over 300 stars). We would include the de-anonymized URL in a camera-ready version.
>
> [11]  CodeT: Code generation with generated tests. (Chen et al. 222)

---

> > ### Comment · Reviewer_fDT5 · 2023-08-17
> >
> > Thanks for the response! I've read the response and the other reviews, this is a nice work and I vote for acceptance.
> >
> > Just one minor question for further discussion: How could Parsel potentially be applied to real-world scenarios, such as programming and robot control?

---

> > > ### Author Response · Authors · 2023-08-18
> > >
> > > Great question!
> > >
> > > We discuss this a bit in Appendix A. For programming, we think there are two main ways in which real-world programmers could use Parsel. First, writing the Parsel sketches from scratch, including some of the associated tests, should allow for programmers to work at a higher level of abstraction while prioritizing test-driven development. People may still look "under-the-hood" at the programs generated, but as long as the language models continue to get more reliable, this should become less necessary. This should support nearly arbitrarily large codebases, and, with good tests and future Parsel features like a more standard object-oriented syntax, could realistically speed up development. With generated tests, it may also facilitate more complete code coverage. The second way is with Parsel generation, for generating longer snippets of code that are beyond the ability of whatever the current best language model happens to be, but likely not in production. Alternatively, with Parsel generation, the initial sketch may be generated by a language model but then revised by programmers.
> > >
> > > For robot control, the Parsel generator can allow end users to describe an often-repeated task and then have the model generate a flexible algorithmic plan for that particular task - one future direction that would benefit this use substantially is mentioned in Appendix C, specifically incorporating more detailed robotic asserts like in [56] and testing potential programs in simulation. Real-time robotic use would likely require further advances in language models, and potentially balancing the compute used for plan generation and implementation generation, since implementing individual functions is often simpler than planning to solve a complex task.
> > >
> > > We also believe Parsel may be a valuable educational tool, as it may allow teaching students algorithmic problem solving removed from the details of programming syntax. Lastly, as we discuss briefly in Appendix D, we also anticipate that Parsel may be useful in formal theorem proving, decomposing proofs into lemmas and then using the formal proof checker as the constraint.
> > >
> > > [56] "Progprompt: Generating situated robot task plans using large language models." Singh et al. 2022

---

> > > > ### Comment · Reviewer_fDT5 · 2023-08-21
> > > >
> > > > Thanks for the response! I have no questions now.

---

### Official Review · Reviewer_PK3i · 2023-07-06

**Soundness:** 4 excellent
**Presentation:** 4 excellent
**Contribution:** 4 excellent
**Rating:** 8
**Confidence:** 4

**Summary:**

This paper proposes Parsel, a code generation framework that decomposes a problem specification into subproblems specified in a intermediate pseudocode-like language (the Parsel language) and then searches over combinations of subproblem solutions. The experiments show that decomposing and searching over subproblems leads to gains on standard code generation benchmarks and a robotic planning task.


**Strengths:**

- Very well-written and motivated
- Thorough experiments on performance gain + analysis of inference cost / scale


**Weaknesses:**

No major weaknesses, besides the minor questions on clarity below.

**Questions:**

- When you set a budget of number of evaluations, how do you decide what to evaluate?
- The "Functions without constraints" section was dense.
	- How does the parent enforce constraints?
- Human expert comparison: this was a cool experiment, but the setup wasn't clear to me: how was the expert using Parsel? Were they initialized with a Parsel-generated ?program and then revised from there, or was it more of an iterative process
- l212-216: I wasn't clear on why this is the conclusion from the human expert results
- nit Fig 4: it initially wasn't clear that 50 (Improved) was the Codex API and that this was the "prior work" you were referring to on line 145
- nit Fig 4: what is 5@50000?
- It'd be helpful to have an example of mutual recursion. I see this is possible in theory but couldn't think of when this would occur and how common it is - why would two function implementations depend on each other?
- How often are child functions shared between parents / called between different parts of the program? In cases where it is shared, it seems like it'd be helpful for the child description to be generated conditioned on _all_ the caller descriptions (whereas from my understanding, the child description is currently generated conditioned on the description of the first caller). I could see this not being an issue for current code generation problems, but it seems like it'd be important in real-world programs.
- Why is it sufficient to only consider strongly connected components?

**Limitations:**

Yes

---

> ### Author Rebuttal · Authors · 2023-08-10
>
> Thank you very much for your comments and questions! We intend to incorporate our responses into the revised paper.
>
> > When you set a budget of number of evaluations, how do you decide what to evaluate?
>
> We consider a random subsample of the possible combinations.
>
> > The "Functions without constraints" section was dense. How does the parent enforce constraints?
>
> The parents’ constraints are used to validate the children. If a child’s parent passes the parent’s constraints, it’s assumed that the functions it depends on are correct (which can, of course, be an incorrect assumption in many situations). In other words, if function $f$ calls function $g$ and $f$ has tests but $g$ doesn’t, then we generate implementations of both $f$ and $g$ and then aim to find a pair of implementations that satisfy $f$’s tests.
>
> > Human expert comparison: this was a cool experiment, but the setup wasn't clear to me: how was the expert using Parsel? Were they initialized with a Parsel-generated ?program and then revised from there, or was it more of an iterative process
>
> They started from scratch, so did not use the Parsel generator. Instead, they solved the problem and wrote their solution in Parsel, which was then synthesized into Python.
>
> > l212-216: I wasn't clear on why this is the conclusion from the human expert results
>
> There were a subset of problems that were hard, such that we were unable to generate Parsel solutions to them by prompting the language model directly. However, when asking a human to generate a Parsel solution, they were able to solve multiple new problems. This suggests that the language is not the primary
>
> > nit Fig 4: it initially wasn't clear that 50 (Improved) was the Codex API and that this was the "prior work" you were referring to on line 145
>
> Thanks for the point! The prior work that that’s from is [11], but we’ll make this more clear.
>
> > nit Fig 4: what is 5@50000?
>
> n@k is how AlphaCode referred to the pass rates, so they used this to refer to selecting 5 from 50,000 samples, and this was their highest-reported number.
>
> > It'd be helpful to have an example of mutual recursion. I see this is possible in theory but couldn't think of when this would occur and how common it is - why would two function implementations depend on each other?
>
> We have one example in Figure 1, e.g., the recursive Collatz conjecture. In practice, this is surprisingly common and almost all cases of recursion can be expressed this way. For example, in our lisp interpreter, get_procedure calls eval_procedure which itself calls get_procedure. Similarly, eval_exp depends on list_case which depends on eval_exp.
>
> > How often are child functions shared between parents / called between different parts of the program? In cases where it is shared, it seems like it'd be helpful for the child description to be generated conditioned on all the caller descriptions (whereas from my understanding, the child description is currently generated conditioned on the description of the first caller). I could see this not being an issue for current code generation problems, but it seems like it'd be important in real-world programs.
>
> Currently, children do not get information about their parents in their implementations. This has the additional benefit of caching: if you change only a function’s description, you do not need to regenerate its children. However, if a function has multiple children (i.e., depends on multiple functions) then all of those functions are listed.
>
> > Why is it sufficient to only consider strongly connected components?
>
> The strongly connected components correspond to the sets of functions whose behaviors are mutually dependent. If two functions depend on one another, they must be implemented together and they will be in the same strongly connected component. If the functions do not depend on one another, then either 1) neither depends on the other or 2) one depends on the other, in which case we can implement the dependency first.

---

> > ### Comment · Reviewer_PK3i · 2023-08-10
> > **Acknowledgment**
> >
> > I've read the rebuttal response — thanks for answering my questions!

---

### Official Review · Reviewer_LfRB · 2023-07-07

**Soundness:** 4 excellent
**Presentation:** 4 excellent
**Contribution:** 3 good
**Rating:** 7
**Confidence:** 4

**Summary:**

This paper introduces Parsel—an intermediate programming language for large language model program synthesis. It decomposes a complex programming task into several strong-connected components and uses large language models to generate candidate code pieces for each component, combined with combinatorial search and test case generation. Evaluation on HumanEval and APPS demonstrate the method's outstanding performance when used with GPT-4 and Codex.

**Strengths:**

- I like the idea of combining language models for function generation and using combinatorial search to find valid programs. It is an approximate form of the fast-and-slow thinking of human problem-solving. The idea is intuitive enough, and the method works well.

- The Parsel language is well-designed as an intermediate programming language in the program synthesis pipeline. It takes advantage of both modeling uncertainty of human language and deterministic descriptive language (IO-specs).

- Experiment results on HumanEval, Apps and VirtualHome are good, reaching outstanding performance with a lower budget (large language model API budget) by trading some local computation cost (combinatorial search), which is something good to see in the LLM era.

- Analysis of Parsel on HumanEval is also valid and persuasive for proving that Parsel can generate longer and more complex programs.


**Weaknesses:**

Overall, I like the paper. I have a few concerns about the weakness.

- (minor) reproducibility: OpenAI is ending their public API service for Codex recently (I think it is before the submission of NeurIPS), resulting recent paper using Codex is hard to reproduce the experimental results after. I suggest using publicly available GPT-3.5 (text-davinci-003 or gpt-3.5-turbo) to give additional reproducible results.

- Parsel design: I am thinking of if the Parsel language is expressive enough (for example, it may lack data structure design (which is something that makes APPS hard). Can Parsel generate data-structure specs?

**Questions:**

See weakness. I have no additional questions.

**Limitations:**

The authors have addressed limitations in the current draft.

---

> ### Author Rebuttal · Authors · 2023-08-10
>
> Thank you for the encouraging review!
>
> > Parsel design: I am thinking of if the Parsel language is expressive enough (for example, it may lack data structure design (which is something that makes APPS hard). Can Parsel generate data-structure specs?
>
> Thank you for this excellent point. This is definitely a challenge. One approach which we’ve used is to manually apply a header (discussed in Appendix Q.2) for this - for example, you might specify that an object is a dictionary with certain keys. As we mention in like 356, for the lisp interpreter, we described the environment dictionary in the header. However, this is clearly an imperfect substitute for a proper object-oriented syntax. We’ll add this to the limitations of Parsel for programmers in Appendix A.1.1 and elaborate it in the main text.
>
> > (minor) reproducibility: OpenAI is ending their public API service for Codex recently (I think it is before the submission of NeurIPS), resulting recent paper using Codex is hard to reproduce the experimental results after. I suggest using publicly available GPT-3.5 (text-davinci-003 or gpt-3.5-turbo) to give additional reproducible results.
>
> At the moment, OpenAI has (fortunately) extended researcher access to Codex. In the long term, we are optimistic that open-source language models (both for code in particular and code and natural language generally) will also become more competitive with these tools. We mention briefly that we attempted to apply these techniques on a 2.7B parameter Codegen model, but even with the same Parsel outlines, it was unsuccessful.

---

> > ### Comment · Reviewer_LfRB · 2023-08-10
> > **Reply to rebuttal**
> >
> > Thank the author for the response! My rating remains the same, and I recommend acceptance for this work.

---

### Author Rebuttal · Authors · 2023-08-10

We thank the reviewers for their thoughtful, positive, and constructive comments! Your suggestions have helped strengthen the work and clarify key points. Based on the reviews, here are some of the main changes:
1. We conducted a new ablation study generating Parsel directly from problems, ablating the high-level plan, in addition to the earlier ablation where we ablated the Parsel synthesizer. This lowered performance to 13% from 25.5% (on a random 200 competition-level APPS problems), showing that both the plan and Parsel decomposition are important.
2. We will clarify details and bring key information which is currently in the appendix into the main text, such as the number of evaluations and the mechanism behind CodeT [11].
3. We will highlight more key takeaways for the sections. For example, for HumanEval, we will note that “Given the same set of generated tests and program generation budget, and selecting only one best solution, Parsel significantly increases the probability that the solution is correct.”

Once again, we really appreciate the supportive feedback and strongly believe that these reviews have strengthened the work.

[11]  CodeT: Code generation with generated tests. (Chen et al. 222)

---

### Decision · Program_Chairs · 2023-09-21

**Decision:**

Accept (spotlight)

**Comment:**

This paper introduces Parsel which is a framework for designing and implementing complex programs. Parsel has an intermediate language that is aimed at decomposing the task into simpler ones at a very high level that includes function description and potential constraints/test cases. This allows sampling and verifying code for individual components separately while handling dependencies. As a result, Parsel enables a clever sampling that makes it more likely for the overall program to be correct. Authors demonstrate this on multiple known benchmarks. I really enjoyed reading the paper and all reviewers unanimously voted for acceptance. Therefore, I recommend acceptance.